# Unraveling the Role of the NLRP3 Inflammasome in Lymphoma: Implications in Pathogenesis and Therapeutic Strategies

**DOI:** 10.3390/ijms25042369

**Published:** 2024-02-17

**Authors:** Ioanna E. Stergiou, Christos Tsironis, Stavros P. Papadakos, Ourania E. Tsitsilonis, Meletios Athanasios Dimopoulos, Stamatios Theocharis

**Affiliations:** 1Department of Pathophysiology, School of Medicine, National and Kapodistrian University of Athens, 11527 Athens, Greece; stergiouioa@med.uoa.gr (I.E.S.); tsironischristos@gmail.com (C.T.); 2First Department of Pathology, School of Medicine, National and Kapodistrian University of Athens, 10679 Athens, Greece; stpap@med.uoa.gr; 3Flow Cytometry Unit, Department of Biology, School of Science, National and Kapodistrian University of Athens, 15784 Athens, Greece; rtsitsil@biol.uoa.gr; 4Department of Clinical Therapeutics, School of Medicine, National and Kapodistrian University of Athens, Alexandra Hospital, 11528 Athens, Greece; mdimop@med.uoa.gr

**Keywords:** NLRP3, inflammasome, pyroptosis, lymphoma, lymphopoiesis, lymphomagenesis, inflammatory signaling

## Abstract

Inflammasomes are multimeric protein complexes, sensors of intracellular danger signals, and crucial components of the innate immune system, with the NLRP3 inflammasome being the best characterized among them. The increasing scientific interest in the mechanisms interconnecting inflammation and tumorigenesis has led to the study of the NLRP3 inflammasome in the setting of various neoplasms. Despite a plethora of data regarding solid tumors, NLRP3 inflammasome’s implication in the pathogenesis of hematological malignancies only recently gained attention. In this review, we investigate its role in normal lymphopoiesis and lymphomagenesis. Considering that lymphomas comprise a heterogeneous group of hematologic neoplasms, both tumor-promoting and tumor-suppressing properties were attributed to the NLRP3 inflammasome, affecting neoplastic cells and immune cells in the tumor microenvironment. NLRP3 inflammasome-related proteins were associated with disease characteristics, response to treatment, and prognosis. Few studies assess the efficacy of NLRP3 inflammasome therapeutic targeting with encouraging results, though most are still at the preclinical level. Further understanding of the mechanisms regulating NLRP3 inflammasome activation during lymphoma development and progression can contribute to the investigation of novel treatment approaches to cover unmet needs in lymphoma therapeutics.

## 1. Introduction

The crosstalk between inflammation, immunity, and cancer represents a field of extensive investigation [1]. Inflammasomes are multimeric protein complexes sensing danger signals implicated in innate immune responses [2,3,4,5]. The best characterized and most well-studied inflammasome is nucleotide-binding oligomerization domain (NOD)-like receptor family pyrin domain containing 3 (NLRP3). NLRP3 inflammasome assembly leads to the activation of caspase-1, thereby promoting the secretion of bioactive interleukin (IL)-1β and IL-18. Overall, NLRP3 inflammasome activation has two main effects: the initiation of a proinflammatory response and/or the induction of an inflammatory type of cell death called pyroptosis.

The function of NLRP3 inflammasome is essential for the maintenance of cellular homeostasis. Its activation is necessary for the efficient control of pathogen infections, but excessive inflammasome activity can result in the development of disease. *NLRP3* mutations have been typically associated with a group of autoinflammatory diseases, the so-called cold-induced autoinflammatory syndrome (CAPS) [6]. The pathophysiology of various other diseases, such as gout, rheumatoid arthritis, inflammatory bowel disease, diabetes mellitus, and Parkinson’s disease, was also shown to include inflammasome dysregulation [7].

Genetic NLRP3 aberrations can affect gene expression and activation. Apart from autoinflammatory diseases, where the significance of polymorphisms is well documented, in a similar way, it was shown that *NLRP3* polymorphisms can be linked to cancer, namely colorectal cancer [8], melanoma [9], pancreatic cancer [10], and gastric cancer [11]. Furthermore, NLRP3 upregulation, identified by gene expression profiling studies, was implicated in a variety of malignancies, such as head and neck squamous cell carcinomas [12], laryngeal squamous cell carcinoma [13], oral cavity squamous cell carcinoma [14], and bladder cancer [15], and was also associated with metastatic properties and prognosis. Despite these reports supporting the pro-tumorigenic role of the NLRP3 inflammasome, we should note that research has additionally indicated a tumor-protective role in certain cancers, such as colitis-associated cancer [16], colorectal cancer [17], hepatocellular carcinoma [18], and melanoma [19]. Therefore, it is believed that the implication of the NLRP3 inflammasome in both tumorigenesis and antitumor immunity is dependent on several factors, such as its levels of expression, its downstream effector molecules (i.e., IL-1β or IL-18), the cancer-type and stage, as well as the potential presence of genetic aberrations affecting its expression or function [20].

Adding to the accumulating data on solid tumors, a recently increasing number of studies investigate the role of the NLRP3 inflammasome in the pathogenesis of hematologic malignancies, with most of them focusing on neoplasms of myeloid origin [21,22,23]. Therefore, in this review, we will try to unravel the less explored involvement of the NLRP3 inflammasome in the development of lymphomas.

Lymphomas comprise a heterogeneous group of hematologic neoplasms, with diverse aetiologies, treatment approaches, and outcomes, that can be roughly subdivided into three major groups: B-cell lymphomas, T and natural killer (NK) cell lymphomas, and Hodgkin lymphomas [24]. According to the latest GLOBOCAN data, 544,352 new cases of non-Hodgkin lymphomas (NHLs) were diagnosed globally in 2018, comprising 2.8% of worldwide cancer diagnoses [25]. Lymphomas of B-cell origin are more common compared to T-cell lymphomas, which account for only 10–15% of NHL diagnoses [26]. The most common NHL in Western countries is diffuse large B-cell lymphoma (DLBCL), accounting for around 31% of adult cases [26]. Despite continuous progress in the therapeutics of lymphoma with the introduction of novel immunotherapies and chimeric antigen receptor (CAR)-T-cell therapy, there is still a subset of patients with dismal prognosis.

Even though NLRP3 inflammasome expression was first described in cells participating in the innate immune response, such as granulocytes, monocytes, macrophages, and dendritic cells [5,27], subsequent research highlighted its role in lymphocyte development and functions [28,29,30,31]. The classification of B-cell lymphomas specifically is based, in part, on the resemblance of a given lymphoma subtype to a particular stage in B-cell development and differentiation, which reflects their origin and instructs their biology [24]. Considering the above, we first try to decipher how the NLRP3 inflammasome participates in the process of normal lymphopoiesis, and we further proceed with its implication in lymphomagenesis. Finally, we review current evidence supporting NLRP3 inflammasome targeting as a potential therapeutic strategy for lymphomas.

## 2. The NLRP3 Inflammasome: Structure, Activation, and Downstream Effects

### 2.1. Structure

The inflammasome is an intracellular multimeric protein complex and a major component of the innate immune system [32]. First described by Jurg Tschopp and his team in 2002 [33], the inflammasome was since found to be implicated in multiple physiological and pathological processes, including antimicrobial defense, cancer, and autoimmunity. As a driver of the inflammatory response, the inflammasome is a pattern-recognition receptor (PRR), mediating cytokine release and pyroptosis, a type of programmed cell death with pro-inflammatory features. Deficient or aberrant inflammasome activation is studied in various disease models [32,34,35].

Though multiple variants of the inflammasome were described, some characterized by a higher degree of complexity, we will focus on the NLRP3 inflammasome, also known as cryopyrin, as a structural prototype. The inflammasome is a multimeric structure consisting of three main components: a sensor protein, an apoptosis-associated speck-like protein containing a CARD (caspase activation and recruitment domain) (ASC) adaptor, and a CARD-containing pro-caspase 1 [36]. In the case of the NRLP3 inflammasome, the sensor subunit is NLRP3. Despite its name, NLRP3 acts mainly as a “sensor” of intracellular damage-associated molecular patterns (DAMPs), namely being activated by them without a direct ligand-to-receptor interaction, which characterizes other PRRs such as toll-like receptors (TLRs). NLRP3 C-terminal is a leucine-rich repeats (LRR) domain, forming an alpha/beta horseshoe fold, a functional configuration acting as a stimulus-sensing structure. The N-terminal is a pyrin domain (PYD) responsible for the sensor–ASC adaptor interaction, while between the two there is a central NAIP (neuronal apoptosis inhibitor protein), C2TA [major histocompatibility complex (MHC) class 2 transcription activator], HET-E (incompatibility locus protein from Podospora anserina), TP1 (telomerase-associated protein) (NACHT) domain. The “caspase subunit” consists of a CARD domain that converts pro-caspase 1 to its active form upon the inflammasome assembly, while the ASC adaptor forms a speck-like structure that binds the sensor to the caspase component through PYD-PYD and CARD-CARD interactions, rendering the inflammasome functional as a whole. Multiple ASC dimers can further polymerize, forming a supramolecular assembly termed the NLRP3 inflammasome or pyroptosome, which is a platform of caspase-1 activation [36,37,38]. A schematic representation of the NLRP3 structure is illustrated in Figure 1.

While NRLP3 was studied more than any inflammasome, other subsets were also described in the literature, each having distinct structural and functional traits, residing in different cell types, and triggered by a wide range of stimuli [2]. To name a few, the NLRP family consists of several inflammasome subsets (NLRP1, NLRP3, NLRP6, NLRP12), all having an NLRP protein as a sensor, yet also found in different cell types other than immune cells, such as glial cells and intestinal epithelial cells, and responding to different DAMPs [39,40]. The NLR family CARD domain-containing protein 4 (NLRC4) inflammasome, which presents a similar structure to NLRP type inflammasomes, has nevertheless a CARD domain integrated into the sensor component and, thus, can be activated without the ASC adaptor recruitment [41]. An increasing amount of evidence demonstrates that a variety of substrates can trigger inflammasome assembly and activation, such as bacterial structural proteins or toxins, as well as other endogenous or exogenous substances and agglomerates, namely monosodium urate crystals, cholesterol crystals, or silica particles [3]. Additional variants present a different type of sensor component, such as the absent in melanoma 2 (AIM2)-like receptor (ALR)-inflammasome, that has a cytoplasmic double-stranded DNA-sensing activity [2,42].

### 2.2. Priming and Activation

Among numerous DAMPs that can either directly or indirectly activate the inflammasome complex, we will describe the principal priming and activation mechanisms of the NLRP3 inflammasome.

For the inflammasome complex to be activated, first, it must undergo a process termed “priming”, involving modifications at the transcriptional and post-transcriptional levels. A “late priming” takes place more than three hours before the inflammasome can be fully recruited and includes the inducible expression of the NLRP3 sensor and pro-IL- 1β. Mediators include molecules in the downstream signaling pathway of TLR activation. Of note, other inflammasome subunits, such as the ASC adaptor or pro-caspase 1, are constitutively expressed by immune cells, including macrophages, and as such, they are not affected by priming in terms of protein expression. “Early” or “intermediate” priming involves post-translational modifications that are carried out in less than one hour since they are not influenced by the time-consuming process of gene expression. Mediators include various stimuli, such as cytokines or chemokines triggering a signaling cascade, that will ultimately result in protein modification via numerous mechanisms, including ubiquitination, deubiquitination, phosphorylation, and ADP-ribosylation [3,4,36,43].

Interestingly, NLRP3 is activated by a multitude of stimuli, as it is a sensor of intracellular signals rather than a ligand-specific receptor [4]. The effect of potassium ion efflux on NLRP3 activation has long been established. An array of mechanisms is known to act on potassium ion efflux, such as extracellular ATP binding to the P2X purinoceptor 7 (P2X7R), which is itself a ligand-gated potassium ion channel contributing to potassium ion efflux [4,44]. Additionally, the influx of calcium ions to the cytosol, originating from either the extracellular space or the endoplasmic reticulum, was also described as an inflammasome activator [4,45]. Other mechanisms of activation involving intracellular electrolyte disturbances, including chloride ion efflux, are also of relevance [4,46]. Moreover, the lysosomal lysis and cytoplasmic release of lysosomal enzymes, such as cathepsins, secondary to the phagocytosis of extracellular agglomerates including monosodium urate crystals, cholesterol crystals, and particles of mineral substances, namely asbestos and silica, further extend the scope of inflammasome activators [4,47,48,49]. Recently, the role of mitochondrial dysfunction on NLRP3 activation through mitochondrial reactive oxygen species (mtROS), mitochondrial DNA (mtDNA), and cardiolipin exposure on the outer mitochondrial membrane was highlighted [4,50,51,52]. According to Chen and Chen, the common denominator among various types of stimuli is the trans-Golgi network (TGN) disassembly in response to those stimuli [53]. In short, disassembled TGN serves as a scaffold for the inflammasome assembly through NLRP3-ASC adaptor interaction, as phosphatidylinositol-4-phosphate on TGN membranes promotes NLRP3 aggregation as well as ASC and caspase-1 recruitment, via a potassium ion efflux-dependent or -independent manner [36,53].

### 2.3. Downstream Response

NLRP3 inflammasome responses are conducted by two mechanisms, termed the canonical and the non-canonical pathways [4,36]. In the canonical response, NLRP3 inflammasome recruitment ends up in caspase-1 activation and the subsequent activating proteolytic cleavage of pro-IL-1β, pro-IL-18, and gasdermin D (GSDMD) into their active forms. Both IL-1β and IL-18 are members of the IL-1 cytokine superfamily [54]. IL-1β is a classic mediator of acute inflammation, produced by a multitude of cells yet predominantly expressed by macrophages. It has long been established that IL-1β is a pyrogenic cytokine, implicated in leukocyte margination and adhesion by inducing E-selectin and adhesion molecule exposure on endothelial cell surfaces, as well as by activating the nuclear factor kappa-light-chain-enhancer of activated B cells (NF-κB) and the mitogen-activated protein kinase (MAPK) pathways [55,56,57]. Moreover, additional data support that the NRLP3 inflammasome is implicated in the Th17 polarization of T cells, possibly through an IL-1β-mediated signaling pathway [58]. IL-18, another inflammatory cytokine, is a major driving factor of T-helper cell differentiation into Th1 and the further activation of macrophages by the Th1/interferon-γ (IFN-γ) response [58,59]. Recent studies support that the inflammasome is also responsible for a Th2-driven humoral/B-cell immune response, yet conclusions remain controversial, as contradictory results regarding the topic were observed [58]. GSDMD is cleaved by caspase-1, so the functional N-terminal fragment is produced (GSDMD-N). GSDMD-N monomers associate with oligomers, forming pores on the cellular membrane, ultimately permitting cytokine escape from the cytoplasm, as well as cell swelling and lysis, due to water entry inside the cell. The combined phenomenon of cell death, IL-1β/IL-18 release, and subsequent inflammatory response is known as pyroptosis. In the case of non-canonical activation, cytoplasmic lipopolysaccharide (LPS) directly stimulates caspase-4 and caspase-5 activation in humans or caspase-11 in mice. The caspase-dependent proteolytic cleavage of GSDMD leads to the formation of GSDMD-N-composed transmembrane pores; nevertheless, no interleukin activation can be accomplished by the non-canonical pathway alone [3,4,36].

Figure 2 summarizes the processes of NLRP3 inflammasome priming and canonical activation, as well as the downstream effects of NLRP3 inflammasome complex assembly.

## 3. The NLRP3 Inflammasome in Lymphocyte Development

Apart from their major role in the innate immune response, recent studies demonstrate the expression and activation of inflammasomes in lymphocytes, highlighting their implication in the adaptive immune response [60]. Describing the latter is beyond the scope of this manuscript and was extensively reviewed elsewhere [61]. NLRP3 expression was reported in the early stages of hematopoiesis, namely in postnatal murine and human hematopoietic stem/progenitor cells (HSPCs) [62], while the participation of inflammasomes in the maintenance of hematopoietic homeostasis was studied mainly in the level of HSPCs [23,63]. Nevertheless, we will focus on the less studied NLRP3 inflammasome implication in the process of lymphopoiesis, considering lymphomas, especially B-cell NHLs, as classified based on the normal counterpart, or cell of origin, from which they arise.

### 3.1. Normal Lymphopoiesis Overview

In adults, lymphocytes originate from bone marrow (BM) hematopoietic stem cells (HSCs) after several stages of differentiation. The pluripotent HSCs initially give rise to multipotent progenitors (MPPs) that will further differentiate into two main progenitor populations: common granulocyte/erythrocyte/megakaryocyte/monocyte (GEMM) progenitors and common lymphocyte progenitors (CLPs). The latter give rise to pre-B-cells, pre-T-cells, and pre-NK-cells. Pre-B-cell differentiation to mature B-cells takes place in the BM, while pre-T-cells leave the BM to complete their differentiation in the thymus [64,65].

B-cell development follows a continuum of stages, beginning in primary lymphoid organs (i.e., the fetal liver, the fetal BM, and adult BM) and culminating in B-cell functional maturation in secondary lymphoid organs [i.e., the lymph nodes (LNs) and the spleen] [66]. The complex of the membrane-bound immunoglobulin (Ig) expressed on B-cells with the CD79 transduction moiety comprises the B-cell receptor (BCR). The generation of a functional BCR determines the survival and maturation of B-cells in the adult BM, as well as the differentiation of mature B-cells that exit the BM and enter the secondary lymphoid organs. This process is regulated by the combinational rearrangement of the Ig gene loci, which also ensures the vast diversity of Igs. Specifically, Ig heavy chain gene rearrangement begins in the early pro-B-cell stage with *D* to *J*, followed by *V* to *DJ* joining, which, if successful, results in the generation of the pre-B-cell. The subsequent *V* to *J* rearrangement of the Ig light chain results in the expression of functional IgM and IgD on the surface of mature B-cells before they exit the marrow. These mature but antigen-naïve B-cells differentiate and gain additional diversity via somatic hypermutation of the *V* genes of the heavy and light chains after exposure to antigens in the germinal centers (GCs) of the secondary lymphoid organs. B-cells expressing Igs with appropriate antigen affinity further differentiate into memory B-cells and plasma cells [66,67,68].

As previously mentioned, in contrast to B-cells, pre-T-cells exit the BM and mature in the thymus. Analogously to B-cells, T-cells recognize specific antigens via the T-cell receptor (TCR). TCR diversity is generated by the recombination of the *V*, *D*, and *J* gene segments of the four TCR genes, alpha (α), beta (β), gamma (γ), and delta (δ). When pre-T-cells reach the thymus, they do not express any of the characteristic T-cell-identifying molecules, namely CD4, CD8, and the TCR complex (i.e., CD3 combined with either α/β or γ/δ TCR chains). These precursors give rise to a minority population of γδΤ-cells (which, even when mature, do not express CD4 or CD8) and the predominant population of αβΤ-cells. The initial population of double negative (DN) for CD4 and CD8 αβT-cells further develops to express both CD4 and CD8, the so-called double positive (DP) αβT-cells. The DP αβΤ-cells whose TCR receptor can interact with self-MHC molecules lose the expression of either CD4 or CD8, finally giving rise to single-positive (SP), CD4, or CD8 mature T-cells that exit the thymus. Of note, αβTCRs can recognize antigens presented only in the context of MHC molecules, while γδTCRs do not have this restriction [68,69].

### 3.2. The NLRP3 Inflammasome in B-Cell Lymphopoiesis

Most data regarding the implication of the NLRP3 inflammasome in B-cell lymphopoiesis are derived from research in animals. NLRP3 inflammasome activation seems to exert a role in the homeostasis between myelopoiesis and lymphopoiesis in early B-cell development in the BM, as well as in B-cell differentiation in the secondary lymphoid tissues.

A decline in B-cell lymphopoiesis and a hematopoiesis shift towards myelopoiesis is a phenomenon observed with aging in humans and mice [70,71]. Aging is also correlated with increased BM fat and an inflammatory environment referred to as inflammaging [72,73]. NLRP3 inflammasome activation was implicated in the inflammatory process mediating impaired B-cell lymphopoiesis. Kennedy and Knight have shown that adipocytes drive the development of myeloid-derived suppressor cells (MDSCs) in vitro, which, in turn, inhibit B-cell lymphopoiesis via IL-1β production [74]. Taking a step further, the same research team proved in a rabbit model, where lymphopoiesis arrests by the age of 2 to 4 months, that these inflammatory changes are driven by the NLRP3 inflammasome. The characterization of the rabbit BM after the arrest of B-cell lymphopoiesis revealed an increase in fat and CD11b+ myeloid cells, along with the upregulated expression of the inflammatory molecules IL-1β and S100A9, produced by myeloid cells. In vitro experiments demonstrated that S100A9 promoted myeloid cells to produce IL-1β, which subsequently inhibited B-cell lymphopoiesis by acting on lymphoid progenitors. NLRP3 inflammasome activation can be mediated by many molecules produced by adipocytes. The NLRP3 inhibitor glibenclamide restored B-cell lymphopoiesis and minimized the induction of myeloid cells induced by the adipocyte-conditioned medium in vitro. It was, therefore, suggested that fat enhances NLRP3 inflammasome activation, which negatively regulates B-cell lymphopoiesis [75].

Furthermore, NLRP3 inflammasome activation was shown to exert a role in B-cell development, homing, and retention in lymphoid organs. Hsu et al. reported that *nlpr3* knock-out in C57BL/6 mice results in perturbations of B-cell development in the BM and the aberrant expression of B-cell subsets with innate-like properties [i.e., marginal zone (MZ) B-cells and B-1a cells] in the periphery. A flow cytometry analysis of BM kappa-positive (κ) and lambda-positive (λ) B-cells revealed a higher representation of κ-positive B-cells in NLRP3-deficient mice compared to wild-type mice [31]. IFN regulatory factor 4 (IRF4) is one of the key transcription factors regulating Ig light chain rearrangement and transcription during pre-B-cell differentiation to the mature B-cell stage [76]. Normally, the rapid expression of IRF4 is triggered by the encounter of immature B-cells with self-antigens, leading to variable gene rearrangements and receptor editing [77]. *Nlrp3* knock-out mice are characterized by increased IRF4 mRNA levels in purified B-cells, a finding suggesting that IRF4 upregulation in the absence of NLRP3 results in κ chain rearrangements at the expense of λ chain assembly [31]. Taking into consideration that NLRP3 interacts with IRF4 in the nucleus of lymphocytes [78], it can be proposed that NLRP3 participates in the regulation of BM B-cell development in an IRF4-dependent manner. In the periphery, *nlrp3* ablation was shown to be responsible for a significant decrease in MZ B-cells in the spleen of the deficient mice, along with an increase in B-1a cells in the peritoneal cavity. The observed altered homing of B-cells can be attributed to impaired chemotaxis since the proportion of BM B-cells expressing both C-X-C chemokine receptor type 4 (CXCR4) and C-C chemokine receptor type 7 (CCR7) were significantly higher in NLRP3 deficient mice compared to controls [31]. These two chemokines are known to regulate B-cell trafficking and homing to secondary lymphoid tissues [79].

In mature B-cells, NLPR3 inflammasome activation either mediates pro-inflammatory cytokine secretion or regulates antibody production depending on the nature of the microbial stimuli. Ali et al. showed that the fungal cell wall carbohydrate β-glucan stimulates Il-1β secretion by B-cells, a process regulated by the NLRP3 inflammasome and dependent on potassium efflux and caspase-1. On the other hand, B-cell activation by unmethylated CpG motifs found in bacterial and fungal DNA failed to induce IL-1β. However, B-cell stimulation by CpG resulted in NLRP3 and caspase-1 activation and the production and secretion of IgM antibodies [80].

### 3.3. The NLRP3 Inflammasome in T-Cell Lymphopoiesis

Age-related thymic involution is characterized by the loss of developing T-cells, a phenomenon contributing to the lower immune surveillance reported in the elderly [81]. Youm et al. described that the age-related increase in lipotoxic danger signals drives thymic caspase-1 activation via the NLPR3 inflammasome. *Nlrp3* or *Asc* knock-out in mice resulted in the reduction of age-related thymic atrophy, as well as in the increase in T-cell progenitors and the maintenance of T-cell repertoire diversity. Moreover, in a mouse model of HSC transplantation after irradiation, *Nlrp3* deletion was shown to accelerate T-cell reconstitution and immune recovery in middle-aged animals [82]. Collectively, these data demonstrate that thymic lymphopoiesis is negatively regulated by NLRP3 inflammasome activation.

## 4. The NLRP3 Inflammasome in Lymphomagenesis

### 4.1. B-Cell NHLs

#### 4.1.1. NLRP3 Inflammasome Activation Contributes to B-Cell NHL Development via Its Effector Cytokines

Regarding DLBCL, NLRP3 inflammasome activation was shown to promote tumor growth and drug resistance. Using the Pfeiffer DLBCL cell line, Zhao et al. demonstrated that NLRP3 inflammasome activation, induced by ATP and LPS treatment, induced lymphoma cell proliferation and inhibited apoptosis through an upregulation in *C-MYC* and *BCL-2* and downregulation in *TP53* and *BAX*. Additionally, it dampened the dexamethasone-induced proliferation-inhibiting effect by promoting cell entry into the S phase. The researchers suggested that the effects of NLRP3 inflammasome activation are mediated by IL-18, given that treatment of Pfeiffer cells with IL-18 also resulted in the proliferation-promoting and apoptosis-inhibiting effects on lymphoma cells via shifting the balance of *C-MYC/TP53* and *BCL-2/BAX*, while neutralizing IL-18 had the opposite effects [83].

Considering that NLRP3 can exert its tumor-promoting effects in DLBCL via its effector cytokine IL-18, we should comment on several studies investigating the prognostic significance of IL-18 expression in this setting. IL-18 mRNA and protein levels are significantly higher in newly diagnosed lymphoma patients compared to healthy controls (*p* = 0.0288 and *p* < 0.0001, respectively), showing a decrease in patients that achieve lymphoma remission after chemotherapy (*p* = 0.0366 and *p* = 0.0098, respectively) [83]. In the pre-Rituximab era, Takubo et al. classified NHL patients based on their IL-18 serum levels before treatment with cyclophosphamide, hydroxydaunorubicin, oncovin, prednisone (CHOP) and compared the outcomes between patients with IL-18 serum levels >2000 pg/mL and those with IL-18 serum levels <1000 pg/mL, reporting statistically significant lower complete remission (CR) rates (33.3% versus 85.7%) and lower median overall survival (OS) (3.5 months versus 45.5 months) for the former group [84]. The impact of pretreatment IL-18 serum levels on the outcome of DLBCL patients receiving Rituximab-CHOP was investigated by Khaled et al. The authors reported that CR and 3-year disease-free survival (DFS) rates were lower for patients with increased IL-18 serum levels, though the difference was not statistically significant. Nevertheless, the 3-year OS rates were significantly more favorable for the group with a lower IL-18 serum level [85]. Using tissue microarrays, Lu et al. highlighted that higher IL-18 expression was detected in DLBCLs with non-germinal center B cell-like (GCB) phenotype compared to those with GCB phenotype [86]. Considering the more aggressive behavior of non-GCB DLBCL [87], we can conclude that higher IL-18 levels characterize more aggressive DLBCL subtypes. We should note, though, that there are also reports in the literature that do not support a prognostic role of IL-18 in B-cell lymphoma patients. Namely, Soydinc et al. measured the serum IL-18 levels in 46 patients with aggressive B-cell NHL (DLBCL and grade III follicular lymphoma) before and after chemotherapy, also using 20 healthy individuals as a control group. They reported no significant difference in IL-18 levels between patients and controls (*p* = 0.261), while they did not find a difference in IL-18 before and after chemotherapy. They reported, though, that serum IL-18 levels were higher in patients with higher lactate dehydrogenase (LDH) levels compared to those with normal LDH levels (*p* = 0.045) [88].

Apart from the most studied DLBCL, NLRP3 inflammasome activation was also implicated in the pathogenesis of mucosa-associated lymphoid tissue (MALT) lymphomas. Baldini et al. proposed the P2X7R-NLRP3 inflammasome axis as a potential pathway implicated in Sjögren syndrome (SS)-related lymphomagenesis. The researchers reported significantly higher mRNA expression for P2X7R, NLRP3, caspase-1, IL-18, and IL-1β in patients with SS who developed MALT-NHL over the follow-up, compared to both SS patients who did not develop lymphoma and controls. Confocal microscopy analysis revealed a more pronounced P2X7R protein expression in SS patients developing MALT-NHL. Of note, the glandular expression of IL-18 was three-fold higher in MALT-NHL than in controls or the other patients with SS. These findings reinforced the hypothesis of a key-role of IL-18 in SS-related lymphomagenesis via an increased P2X7R-NLRP3-mediated production [89].

Research also highlighted the role of the NLRP3 inflammasome in other less frequent B-cell NHLs. Using high-density microarray chips, differential gene expression analysis in mantle cell lymphoma (MCL) demonstrated an upregulation of the *IL-18* gene [90]. Primary effusion lymphoma (PEL) cells show evidence of NRLP3 inflammasome activation, such as the presence of active caspase-1 and cleavage of pro-IL-1β and pro-IL-18 [91].

Interestingly, for some B-cell NHL subtypes, NLRP3 inflammasome activation was reported to act as a negative regulator of lymphomagenesis. Namely, in chronic lymphocytic leukemia (CLL)/small lymphocytic lymphoma (SLL), Salaro et al. highlighted that NLRP3 overexpression by the neoplastic lymphocytes correlated with the inhibition of cell proliferation and induction of apoptosis, while NLRP3 downregulation contributed to lymphomagenesis [92]. NLRP3 was also shown to exert an antitumorigenic effect in Burkitt lymphoma triggered by Epstein–Barr virus (EBV). Reinhart et al. demonstrated that NLRP3 cooperated with high mobility group box 1 (HMGB1) protein to maintain the EBV lytic switch protein ZEBRA expression in Burkitt lymphoma-derived cells, thus sustaining the lytic signal [93].

#### 4.1.2. The Effect of NLRP3 Inflammasome Activation in the B-Cell NHL Microenvironment

Exploring the impact of NLRP3 inflammasome activation on the lymphoma microenvironment, Lu et al. demonstrated that IL-18 levels positively correlated with programmed death-ligand 1 (PD-L1) expression. To prove that this finding reflects the downstream effect of NLRP3 inflammasome activation, the researchers proceeded with NLRP3 inflammasome activation by LPS and ATP stimulation of DLBCL cell lines. This activation led to a PD-L1 upregulation and T-cell decrease. To validate the role of the NLRP3 inflammasome in B-cell lymphoma progression in vivo, a murine model was used. NLRP3 inflammasome blockade by treatment of mice with the inhibitor MCC950 suppressed lymphoma growth and ameliorated antitumor immunity by downregulating PD-L1 in the tumor microenvironment (TME) and decreasing the proportion of programmed cell death protein 1 (PD-1)/T-cell immunoglobulin and mucin-domain containing three (TIM-3)-expressing T-cells, MDSCs, tumor-associated macrophages (TAMs), and regulatory T-cells [86]. Deciphering the pathways implicated in PD-L1 upregulation induced by NLRP3-dependent IL-18 production is of great interest. IL-18 is a potent inducer of IFN-γ [94]. After IFN-γ binding to its receptor, the activation of the Janus kinase (JAK)/ signal transducer and activator of transcription (STAT) signaling pathway leads to IRFs’ induction, which, in turn, mediates PD-L1 expression [95]. JAK/STAT3 signaling was shown to be an up-stream regulator of PD-L1 expression in various tumor models [96,97,98,99]. In the DLBCL study of Lu et al., pSTAT3 protein levels decreased after IL-18 and NLPR3 inflammasome inhibition, possibly linking it to the observed concomitant PD-L1 downregulation [86]. More recently, Serna et al. applied bioinformatics analysis in a TCGA (The Cancer Genome Atlas) dataset, including 48 tissue samples of DLBCL patients, and observed a significant upregulation of key inflammasome components (*AIM2*, *ALK*, *IRF3*, *4* and *8*, *NFKB1* and *2*, *NOD2*, *NLRP1* and *3*, *CASP1* and *5*, *CARD8* and *9*) in M0 and M1 macrophages [100]. It can, therefore, be proposed that NLPR3 inflammasome activation can drive the polarization of macrophages, inducing a pro-inflammatory microenvironment that subsequently shapes the tumor-suppressive properties of other immune cells or contributes to B-cell malignant transformation.

#### 4.1.3. Deciphering the Drivers of NLRP3 Inflammasome Activation in B-Cell NHL

The described NLRP3 inflammasome activation in B-cell NHLs can be, so far, attributed to three parameters: (a) genetic alterations in genes encoding for NLRP3 inflammasome-associated molecules, which favor the upregulation of the NLRP3 inflammasome pathway; (b) aberrations in factors regulating B-cell development that induce the NLRP3 inflammasome pathway; and (c) persistent inflammatory signaling and NLRP3 inflammasome activation in immune cells of the lymphoma microenvironment favoring B-cell malignant transformation.

Research highlighted that certain polymorphisms in the genes encoding for NLRP3 inflammasome-related molecules are associated with increased susceptibility to B-cell NHL development and correlate with lymphoma patients’ survival and prognosis. Liu et al. investigated genetic polymorphisms in NLRP3 inflammasome-related genes in 281 patients with B-cell NHL and 385 age- and gender-matched healthy controls. The study showed that IL-18 (rs1946518) and NF-κB-94 ins/del (rs28362491) gene polymorphisms contributed to a significantly increased risk for B-cell NHL (*p* < 0.0001 and *p* = 0.0029, respectively). Specifically, the alleles responsible for this increased susceptibility to lymphoma development were the “G” in IL-18 (rs1946518) and the allele “ins” in NFκB-94 ins/del (rs28362491). While the distribution of CARD8 rs2043211 polymorphism showed no difference between lymphoma patients and healthy controls, the AA genotype correlated with a statistically significant poorer survival (*p* = 0.0381). Associations between polymorphisms and disease characteristics were also underscored. Namely, the TT genotype of CARD8 (rs2043211) was observed in patients with higher LDH levels, clinical stages III-IV, and international prognostic index (IPI) scores 3–5, although the relationship did not reach statistical significance [101]. Wang et al. analyzed mutations and copy number variations (CNVs) of pyroptosis-related genes (PRGs) in DLBCL, unveiling multiple aberrations that also correlated with disease prognosis. Three distinct pyroptosis-related clusters were identified, characterized by significant differences regarding prognosis, biological processes, clinical characteristics, chemotherapeutic drug sensitivity, and the TME. Most genes were upregulated in DLBCL, and among the upregulated PRGs, *PYCARD*, *IRF1*, *GZMA*, *GSDMD*, *GSDMC*, *GPX4*, *CHMP2A*, *CASP5*, *CASP1*, *Il-18*, and *BAK1* showed an amplified CNV status, suggesting a significant relationship between the PRG expression and CNVs [102]. Likewise, Ma et al. proposed a novel and validated an eight-PRG-based risk prediction model for DLBCL [103].

B-cell activating factor (BAFF) is a member of the tumor necrosis factor (TNF) family ligands crucial to B-cell development and homeostasis [104,105]. Circulating BAFF protein levels are increased in patients with B-cell NHLs compared to healthy controls [106,107]. Several studies identified BAFF as a factor promoting the proliferation and contributing to the apoptosis resistance of the malignant B-cell clone in various hematologic malignancies of B-cell origin [108,109,110,111]. The BAFF/BAFF receptor (BAFF-R) axis is crucial to the survival and growth of malignant B-cells [112]. Apart from the well-studied activation of the NF-κΒ pathway by BAFF signaling, Lim et al. demonstrated that the BAFF/BAFF-R axis can induce NLRP3 priming and activation signals in primary B-cells and lymphoma cell lines. The treatment of normal and neoplastic B-cells with BAFF not only led to an increase in *NLRP3* mRNA expression but also in increased NLRP3 activation (as proven by the increase in active caspase-1 and IL-1β levels) in a time- and dose-dependent manner. The BAFF-induced NLRP3 inflammasome activation was shown to be mediated by the association of the cellular inhibitor of apoptosis protein (cIAP)—TNF receptor-associated factor (TRAF)2 (which is released in the cytoplasm after TRAF3 recruitment to BAFF-R induced by BAFF binding) to NLRP3–ASC–procaspase-1 complexes. Another contributor of BAFF-induced NLRP3 activation was reported to be the induction of Src activity-dependent ROS production and potassium ion efflux. Of note, BCR stimulation on the Lyn signaling pathway inhibited BAFF-induced Src activities and attenuated BAFF-induced NLRP3 inflammasome activation [113].

BAFF upregulation in patients with SS was correlated with increased risk for lymphoma development, predominantly of the MALT lymphoma subtype [114]. Surprisingly, for SS-related lymphomas, research focused on stimulators of NLRP3 activation other than BAFF, acting on non-neoplastic immune cells of the TME. SS patients demonstrate increased amounts of circulating nucleosomes and cell-free (cf) DNA in their blood due to impaired DNaseI-mediated degradation [115,116]. These nucleic acids can act as inducers of inflammasome activation [117]. Vakrakou et al. demonstrated that systemic activation of the NLRP3 inflammasome characterizes SS patients at high risk for lymphoma development and those with established lymphoma, along with significantly increased serum IL-18 and ASC levels. In these patients, the circulating monocytes manifested NLRP3 inflammasome activation and increased response to NLRP3 stimuli, whereas salivary gland (SG)-infiltrating macrophages exhibited signs of NLRP3 activation and pyroptosis. These patients were also characterized by high serum cfDNA levels and substantial extranuclear DNA accumulation in peripheral blood mononuclear cells (PBMCs) and SG tissues. The competent activation of NLRP3 inflammasome in healthy monocytes treated with cf nucleic acids isolated from patients’ sera established the causal relationship for the findings in SS patients [118]. These observations prompt us to highlight the significance of NLRP3 inflammasome activation in immune cells of the lymphoma microenvironment, which, by persistent inflammatory signaling, can interact with B cells, favoring their monoclonal expansion and malignant transformation.

Figure 3 summarizes the proposed mechanisms by which inflammasome activation could participate in the pathogenesis and progression of B-cell NHLs.

### 4.2. T and NK-Cell Lymphomas

As opposed to B-cell NHLs, there are fewer studies investigating the role of the NLRP3 inflammasome in the pathogenesis of T and NK-cell lymphomas, which is not unexpected given the lower frequency of these lymphoma subtypes.

Cutaneous T-cell lymphomas (CTCLs) are extranodal T-cell lymphomas in which the malignant clone arises from skin-homing Th2 cells [119]. Yamanaka et al. proposed that increased IL-18 levels could be held responsible for the observed Th2 bias in this disease. The researchers detected significantly increased plasma levels of IL-18 and caspase-1 compared to healthy controls, while their corresponding mRNA levels were also elevated in skin lesions. We should note, though, that their study design did not include experimental procedures to prove that the observed IL-18 and caspase-1 increase is exclusively driven by NLRP3 inflammasome activation. After the development of three-dimensional skin explant cultures, matrices populated with skin derived from CTCL lesions produced higher amounts of IL-18 and caspase-1 compared to those populated with normal skin cells, while IL-12 production was barely detectable [120]. It was previously demonstrated that IL-18 in combination with IL-12 induces IFNγ production by CD4+ Τ-cells [121], while high levels of IL-18 alone induce the production of IL-4 and IL-13 by basophils, mast cells, and CD4+ T-cells [122,123]. It can, therefore, be hypothesized that in the case of CTCL, high IL-18 levels in combination with almost absent IL-12 can favor the skewing of CD4+ T-cells to the Th2 phenotype, which, in turn, can give rise to the malignant clone.

More recently, Huanosta-Murillo et al. demonstrated that in CTCL, NLRP3 can promote the characteristic Th2 response by controlling IL-4 production. The researchers reported that unassembled NLRP3 can translocate to the nucleus of the malignant CD4+ T-cells, where it binds to the human IL-4 promoter, increasing IL-4 production. This creates a positive feedback loop in which IL-4 inhibits NLRP3 inflammasome assembly, leading to a further increase in IL-4 production. Comparing IL-4 expression in thymocyte selection-associated HMG BOX (TOX)+ CD4+ T-cells between plaque and tumor lesions, a significant increase in IL-4 and TOX expression was observed in the tumor stage, with a positive correlation between the levels of TOX and IL-4 expression. Therefore, IL-4 production mediated by NLRP3 increases with lesion severity and is associated with disease progression [124]. In this case, it seems that the inhibition of NLRP3 inflammasome assembly supports the malignant dynamic of CD4+ T-cells.

Sézary syndrome is a rare leukemic type of CTCL, where the neoplastic cells are characterized as central memory CD4+ T-cells with skin-homing properties [125,126]. Manfere et al. highlighted a compartmentalized expression of NLRP3-related components and provided evidence of their imbalance in patients with Sézary syndrome. Assessing the expression of NLRP1, NLRP3, AIM2, IL-1β, and IL-18 by immunohistochemistry in the epidermal and dermal layers of skin from Sézary syndrome patients, idiopathic erythroderma patients, and healthy donors, the researchers identified increased IL-1β and low IL-18 levels in the epidermal skin layers of Sézary syndrome patients compared to the control groups. However, in the dermal layer, IL-18 showed increased expression in Sézary syndrome patients, while IL-1β levels were comparable among patients and controls. Despite these abnormal patterns of expression, NLRP3 and AIM2 were equally expressed in the different groups, while NLRP1 expression was observed in Sézary syndrome skin compared to controls. Increased serum levels of IL-18 were also detected in Sézary syndrome patients, while the LNs of patients with advanced-stage disease showed an upregulation of IL-18 and downregulation of IL-1β. At the transcriptional level, Sézary syndrome LNs showed no difference in the IL-18 mRNA levels compared to LNs from healthy donors and idiopathic erythroderma patients, while IL-1β and NLRP3 mRNA levels were found to be downregulated [127].

In NK/T-cell lymphoma, high IL-18 serum levels were associated with stage III/IV disease, the presence of hemophagocytosis, and poor treatment outcomes. OS and progression-free survival (PFS) were significantly lower for the high IL-18 group compared to the low IL-18 groups (*p*  < 0.001), and high serum IL-18 was independently prognostic for survival in multivariate analysis [128]. Further research demonstrated that Hsa-miR-372-5p regulates the never-in-mitosis gene A (NIMA)-related kinase 7 and IL-1β release in NK/T-cell lymphoma. It was proposed that Hsa-miR-372-5p may target NIMA-related kinase 7 to regulate NLRP3 inflammasome activation [129].

Table 1 summarizes research findings regarding NLRP3 inflammasome implications in the pathogenesis of various lymphoma subtypes.

## 5. Exploring the Crosstalk of Tripartite Motif (TRIM) Family Proteins and NLRP3 Inflammasome Activation in Lymphoma

Ubiquitination represents one of the main mechanisms of post-transcriptional protein modification. The process of ubiquitination comprises the connection of the ubiquitin C-terminal lysine to the target protein lysine residue via an isopeptide bond through the activity of ubiquitin activator (E1), ubiquitin-binding enzyme (E2), and ubiquitin ligase (E3), culminating in structural and functional alterations of the target protein [130]. The two most extensively characterized modes of ubiquitination are of Lys48 connection, which is recognized by the proteasome leading to protein degradation, and Lys63 connection, modulating signaling pathways, such as those of DNA repair, innate immune response, and NF-κΒ [131,132]. The specificity of ubiquitination is regulated by E3 ligases, the enzymes responsible for identifying the specific target protein [133]. TRIM proteins constitute one of the largest subfamilies of E3 ubiquitin ligases, which was shown to participate in NLRP3 inflammasome regulation at multiple levels via the ubiquitination of various substrates [134], while their dysregulation was associated with hematological malignancies [135], highlighting another possible link between NLRP3-related aberrations and lymphomagenesis. Therefore, in this section, we first briefly analyze the regulation of the NLRP3 inflammasome components and related pathways by TRIMs, and we continue by reviewing research results implicating TRIMs in lymphoma development.

The TRIM regulation of the NLRP3 inflammasome may present both promoting and inhibiting functions. Xie et al. reported that TRIM14 can activate the NF-κB/NLRP3 inflammasome pathway via NF-κB targeting [136]. TRIM16 was shown to interact with pro-IL-1β; pro-caspase-1; and NACHT, LRR, and PYD domains containing protein 1 (NALP1) to enhance IL-1β secretion [137]. On the other hand, Jena et al. reported that TRIM16 induces the activation of nuclear factor erythroid 2–related factor 2 (Nrf2), therefore downregulating ROS production, an effect that can lead to attenuated NLRP3 inflammasome activation [138]. TRIM20 can interact with NLRP3 mediating NLRP3 degradation by Unc-51-like kinase 1 (ULK1) and Beclin1 [139], while it can also downregulate pro-caspase-1 [140]. Conversely, the ability of TRIM20 to bind to the PYD domain of ASC can result in NLRP3 inflammasome activation [141]. TRIM21 was identified as a positive regulator of GSDMD [142]. TRIM22 was shown to favor NLRP3 inflammasome activation via NF-κB [143], while TRIM24 interaction with NLRP3 exerts an inhibitory effect [144]. TRIM25 indirectly dampens NLRP3 inflammasome activation since it induces the ubiquitination and degradation of Kelch-like ECH-associated protein 1 (Keap1), enhancing the nuclear localization of Nrf2 with subsequent inhibition of ROS production [145]. TRIM28 stimulates NLRP3 SUMOylation, promoting its stability and facilitating further activation [146]. TRIM30 downregulates ROS levels, consequently mitigating NLRP3 inflammasome activation [147], while in opposition, TRIM31 contributes to NLRP3 inflammasome activation via its potential to promote TP53-inducible glycolysis and apoptosis regulator (TIGAR) degradation, thus hindering ROS clearance [148]. Of note, TRIM31 was also reported to contribute to NLRP3 degradation via K48- polyubiquitination [149]. DHX3 activation via TRIM33-mediated ubiquitination induces NLRP3 inflammasome activation [150,151]. TRIM40 interaction with NLRP3 results in NLRP3 inflammasome inactivation [152]. TRIM59 drives the abhydrolase domain containing 5 (ABHD5) ubiquitination and successive degradation, leading to increased lactate secretion and ROS-dependent inflammasome activation [153]. TRIM62 upregulates the NF-κB/NLRP3 signaling pathway, enhancing IL-1β and IL-18 transcriptional activity [154], while it can also promote the CARD9/NF-κB/NLRP3 axis via CARD9 ubiquitination and activation [155]. TRIM65 mediates NLRP3 ubiquitination, which results in the downregulation of NLRP3 inflammasome activation [156].

The effects of TRIM proteins’ aberrations were investigated in various hematological neoplasms, both of myeloid and lymphoid origin [135]. In this setting, the most studied TRIM protein is TRIM19, also known as the promyelocytic leukemia (PML) protein. In normal conditions, PML is indispensable for the formation of PML nuclear bodies (PML-NB), which are distinct nuclear structures activated by cellular stress stimuli and participate in the regulation of various cellular processes, such as transcription, cell cycle, apoptosis, senescence, DNA damage response, and anti-viral response [157]. Apart from its extensive study in acute promyelocytic leukemia (APL), which is characterized by the balanced reciprocal chromosomal translocation, t(15;17), producing the PML–retinoic acid receptor α (RARα) fusion protein [158], PML was also implicated in lymphomagenesis, exerting a tumor-suppressor function. Gurrieri et al. reported reduced PML expression in NHLs [159]. The E3 ubiquitin ligase E6AP holds a pivotal role in the regulation of PML stability and the formation of PML-NBs [160]. Wolyniec et al. demonstrated that partial the loss of E6AP attenuated Myc-induced B-cell lymphomagenesis via induction of cellular senescence. B-cell lymphomas deficient for E6AP expressed elevated levels of PML and PML-NBs with a concomitant increase in markers of cellular senescence, while PML deficiency accelerated the rate of Myc-induced B-cell lymphomagenesis. Dong et al. reported that TRIM11 was downregulated in MCL tissues collected from patients and human MCL cell lines, while the long non-coding RNA LUADT1 and miR-24-3p were upregulated compared to hyperplastic lymphadenitis and PB lymphocytes. The researchers demonstrated that LUADT1 could modulate TRIM11 by sponging miR-24-3p to inhibit cancer cell apoptosis [161]. Conversely, Hou et al. documented increased TRIM11 expression in lymphoma tissues, proposing that TRIM11 contributes to lymphomagenesis by activating the β-catenin signaling and inducing ubiquitination-mediated Axin1 degradation [162]. TRIM13 was initially identified as a tumor suppressor gene in B-cell CLL [163], though subsequent studies questioned this observation [164,165]. Nevertheless, TRIM13 expression was found to be downregulated in advanced-stage CLL compared to diagnosis, a finding suggestive of its tumor-suppressive function [166]. TRIM28 was shown to be upregulated in CTCL and B-cell-NHL patient samples [167,168]. For the case of B-cell NHLs, TRIM28 expression was positively and independently correlated with dismal patient survival, while in terms of pathophysiology, TRIM28 was demonstrated to contribute to B-cell NHL proliferation by increasing cyclinA and proliferating cell nuclear antigen (PCNA) expression and reducing P21 expression. Furthermore, the inhibition of TRIM28 expression in B-NHL cells enhanced the sensibility to Bortezomib by regulating the p53-mediated apoptosis pathway [168]. TRIM32 was identified as a target gene of miR-155, whose levels are highly elevated in most B-cell lymphomas [169]. Tan et al. reported reduced TRIM35 expression in human DLBCL tissues, also highlighting that TRIM35 overexpression suppresses DLBCL cell proliferation, a process mediated by the ubiquitination and degradation of CLOCK [170]. TRIM65 overexpression was proposed to contribute to lymphomagenesis via the ERK1/2 signaling pathway, based on studies of Raji cell lines, where TRIM65 overexpression enhanced cell viability and increased the protein levels of Bcl2, VEGF, *p*-ERK1/2 [171].

Notwithstanding, TRIM14, TRIM16, TRIM20, TRIM21, TRIM22, TRIM24, TRIM25, TRIM30, TRIM31, TRIM33, TRIM40, TRIM59, and TRIM62 were implicated in NLRP3 inflammasome regulation [136,137,138,139,140,141,142,143,144,145,147,148,149,150,151,152,153,154,155], and current research has not addressed the question of their possible role in lymphomagenesis. Conversely, despite the suggested involvement in lymphoma development and the progression of TRIM11, TRIM13, TRIM19, TRIM32, TRIM35, TRIM65 [159,160,161,162,163,166,169,170], data on their possible function regarding NLRP3 inflammasome regulation are lacking. The only TRIM family members explored both in the context of NLRP3 inflammasome regulation and their implications in lymphomagenesis are TRIM28 and TRIM65. The positive correlation of TRIM28 expression in B-cell NHL tissues with inferior patient survival [168] could, therefore, be attributed to both its contribution to NLRP3 inflammasome activation [146] and the subsequent enhancement of lymphoma cell survival and proliferation (via pathways that will be analyzed in the following section), and its involvement in the regulation of cell cycle and p53-mediated apoptosis [168]. Interestingly, TRIM65 was shown to downregulate NLRP3 inflammasome activation [156], while in the setting of lymphoma, TRIM65 overexpression favors lymphoma cell survival via the ERK1/2 signaling pathway [171]. Further investigations are undoubtedly needed to unravel the intricate intertwining of TRIM protein-mediated pathways implicated in NLRP3 inflammasome regulation and lymphomagenesis.

## 6. Molecular Pathways by Which NLRP3 Inflammasome Affects Lymphoma Cells and the Immune TME

It was shown that inflammasomes can be activated both in neoplastic cells and immune cells of the TME, while the release of its effector cytokines can exert a tumor-promoting function in a bidirectional way by enhancing the survival and/or proliferation of neoplastic cells and by favoring a tumor permissive immune TME [172,173]. In the following section, we review the effects of NLRP3 inflammasome activation on lymphoma cells and on immune cells of the TME.

### 6.1. Effects on Lymphoma Cells

The IL-18 receptor (IL-18R), consisting of two subunits (i.e., the IL-18Rα chain and the IL-18Rβ chain), is typically expressed in T-cells and NK-cells [174]. Nevertheless, in the case of lymphoproliferative disorders, heterogeneous IL-18R patterns of expression were documented in B-cells [175]. Considering this observation, it can be proposed that IL-18, secreted as a result of NLRP3 inflammasome activation in the setting of lymphoma, can induce signaling pathways promoting lymphomagenesis via its interaction with the IL-18R on lymphoma B-cells. Specifically, the downstream pathways induced upon IL-18 binding to IL-18R include those of NF-κB, phosphoinositide 3-kinase (PI3K)/Akt, and RAS/MAPK [94], all of which were shown to hold an integral role in the survival and proliferation of neoplastic cells in various lymphoma subtypes [176,177,178]. Among genes that are upregulated due to the constitutive activation of the above-mentioned pathways, we should note *BCL2* induced by MAPK/activating transcription factor 2 (ATF-2) signaling [179], as well as *C-MYC* and *CCND1* induced by NF-κΒ signaling [180,181]. These were implicated in the pathogenesis of various lymphoma subtypes via the enhancement of lymphoma cell proliferation and reduced apoptosis [182,183,184]. We should note that analogously, IL-1β, the other downstream effector cytokine of NLRP3 activation, can also lead to NF-κB and MAPK/activator protein 1 (AP-1)-mediated gene transcription [57]. At the same time, it can also promote the β-catenin signaling pathway, leading to the further upregulation of *C-MYC* and *CCND1* expressions [185]. Figure 4A illustrates the molecular pathways implicated in lymphoma cell survival and/or proliferation induced by NLRP3 inflammasome-related effector cytokines.

### 6.2. Effects on the Immune TME

NLRP3 inflammasome activation in lymphoma cells tends to generate an immune-suppressive TME. IL-18 was shown to promote MDSC accumulation in the tumor microenvironment, leading to the suppression of CD4+ and CD8+ T cells, both in solid tumors and hematologic malignancies [186]. Though direct correlations between IL-l8 levels and MDSC infiltration for lymphoma were not reported, based on the study of Lu et al. in which NLRP3 inflammasome inhibition in a murine model leads to the decrease in MDSCs [86], it can be proposed that IL-18 mediates MDSC aggregation in lymphoma cases too. IL-1β was also shown to induce MDSC recruitment and activation in vivo and in vitro in gastric cancer mouse models via the IL-1 receptor (IL-1R)/NF-κB axis [187]. It is noteworthy that for the case of melanoma it was shown that NLRP3 inflammasome activation elicits autocrine heat shock protein 70 (HSP70)/TLR4 signaling pathway inducing a Wnt family member 5A (Wnt5a)-dependent C-X-C motif chemokine ligand 5 (CXCL5)/C-X-C motif chemokine receptor 2 (CXCR2) secretion that recruits granulocytic MDSC [188].

Despite the fact that, in general, NLRP3 inflammasome activation in immune cells exerts antitumor effects, triggering the immune response against cancer cells [35], NLRP3 inflammasome activation in tumor-associated macrophages (TAMs) and the subsequent IL-1β production was shown to promote PD-L1 expression in neoplastic cells via the activation of the NF-κB signaling cascade [189]. Furthermore, IL-1β secretion by TAMs drives an immunosuppressive CD4+ T-cell polarization [190].

The activation of NLRP3 inflammasome in human DLBCL cell lines and its correlation with PD-L1 expression highlights its role in the shaping of a tumor-promoting immune tumor microenvironment [86]. This can be mediated by IFN-γ. Considering that either NLRP3 or IL-18 inhibition in lymphoma mouse models leads to a decrease in PD-1/TIM-3-expressing T-cells, MDSCs, tumor-associated macrophages, and regulatory T-cells, it can be proposed that IL-18 mediates the accumulation of immunosuppressive cell populations [86].

The effects related to NLRP3 inflammasome-induced cytokines on the immune cells of the TME are summarized in Figure 4B.

## 7. NLRP3 Inflammasome Signaling in Lymphoid Premalignant Conditions

Of note, ongoing research suggests the implication of NLRP3 inflammasome in lymphoid premalignant conditions, such as monoclonal B-cell lymphocytosis (MBL) and monoclonal gammopathy of undetermined significance, known to confer increased risk for the development of overt malignancy [191,192]. Blanco et al. demonstrated that PB monocytes of individuals with MBL are characterized by increased inflammatory signaling. NLRP3 inflammasome-related genes (i.e., *NLRP3*, *IL-18*, *IL-1B*) were found to be upregulated, while increased IL-18 serum levels were also reported. Interestingly, this inflammatory signaling decreased in early-stage CLL [193]. These findings are indicative of a distinct inflammatory environment between MBL and CLL, potentially contributing to malignant B-cell transformation. Considering that Il-1β serum levels are increased in only about 10% of individuals with monoclonal gammopathy of undetermined significance (MGUS) [194] and given the role of IL-1β in the pathogenesis of multiple myeloma (MM) [195], it can be proposed that NLPR3-driven IL-1β production might drive the neoplastic phenotype in a subset of MGUS cases. Schnitzler’s syndrome is an example of a disease entity with IgM monoclonal gammopathy as a defining feature and increased risk of evolution to lymphoproliferative disorders (i.e., MM, marginal zone lymphoma, Waldenström’s macroglobulinemia) [196]. Activating *NLRP3* mutations were identified in patients with Schnitzler’s syndrome [197], underscoring another possible link between NLRP3 inflammasome activation and MGUS progression to lymphoid neoplasia. Though the clonal hematopoiesis of indeterminate potential (CHIP) has traditionally been correlated with the evolution to neoplasms of myeloid origin [198], we should highlight that clonal hematopoiesis can potentially give rise to any hematologic malignancy, either myeloid or lymphoid, considering that related mutations can affect less committed hematopoietic precursors. Genes encoding the epigenetic regulators TET2 and DNMT3A, commonly mutated in CHIP, were implicated in inflammasome activation in myeloid cells, while for lymphoid cells, the same mutations were shown to induce methylation-dependent alterations in differentiation and function [199]. Overall, identifying a dysregulated NLRP3 inflammasome signature in premalignant hematological entities could serve both as a biomarker of progression to neoplasia, as well as a possible target to hinder the malignant transformation.

## 8. Therapeutic Targeting of NLRP3 in Lymphoma

The therapeutic targeting of the NLPR3 inflammasome pathway, and mostly drugs suppressing its activation, were predominantly studied in solid tumors [200]. Compared to these reports, there is a relative paucity of data regarding NLPR3 inflammasome targeting in lymphomas. More recent research, though, provides encouraging preliminary results regarding lymphomas of both B- and T-cell origin.

Luxeptinib (CG-806) is an orally bioavailable multikinase inhibitor with nanomolar potency against Bruton’s tyrosine kinase (BTK) and FMS-like tyrosine kinase 3 (FLT3), among other kinases. Ongoing clinical trials assess its efficacy in acute myeloid leukemia (AML) and lymphoma patients [201]. In lymphoma cell lines, Luxeptinib interferes with LYN-mediated activation of SYK, a necessary regulator of BTK, thus modulating BCR signaling [202], while it was also shown to downregulate proteins of the anti-apoptotic Bcl-2 family [203]. Apart from its effect on BCR signaling and its ability to induce apoptosis in lymphoma cells, Luxeptinib was also shown to inhibit NLRP3-mediated IL-1β secretion in monocytic cell lines. Of note, Luxeptinib does not inhibit the assembly of the NLRP3 inflammasome but disables its ability to cleave and activate caspase-1, which is required for IL-1β release [204]. Simultaneous inhibition of B-cell proliferation pathways and inflammatory signaling pathways may provide better therapeutic efficacy. In the setting of B-cell malignancies, the ability of Luxeptinib to concurrently downregulate BCR, Bcl-2, and NLRP3 signaling renders it a possible, more effective therapeutic alternative to the widely used BTK and Bcl-2 inhibitors. As a matter of course, this remains to be further elucidated and evaluated in the setting of clinical trials.

Hematopoietic progenitor kinase 1 (HPK1) is a negative regulator of TCR signaling and, thus, a contributor to reduced antitumor immunity [205]. Based on the Gene Expression Profile Interactive Analysis (GEPIA) database, HPK1 expression was increased in DLBCL and associated with NLRP3 expression [206]. Given that HPK1 inhibition was shown to enhance antitumor immunity [207], Yang et al. explored whether an HPK1 inhibitor could act synergistically with anti-PD-1 immunotherapy for B-cell NHL in terms of tumor response, as well as its effect on NLRP3 expression. They reported that the HPK1 inhibitor increased anti-PD-1-mediated T-cell cytotoxicity in B-cell NHL lines cocultured with PBMCs. HPK1 inhibitor treatment increased PD-1, PD-L1, Bax, p53, and NK-κB expression but decreased NLRP3 expression. These findings indicate that the promotion of apoptosis and NLRP3 inhibition induced by HPK1 inhibition could affect anti-PD-1-mediated T-cell cytotoxicity, providing a potential therapeutic option for B-cell NHL, namely the combination of HPK1 inhibitor treatment and anti-PD-1 immunotherapy [206].

Hasegawa et al. provide the first report showing that histone deacetylase inhibitors (HDACis) are responsible for NLRP3 inflammasome-mediated cell death in adult T-cell leukemia/lymphoma (ATL) cell lines and primary cells from patients. HBI-8000 is a novel oral HDACi with proven efficacy for the treatment of T-cell lymphomas that recently received approval in China. HBI-8000 induced apoptosis in ATL cells, driven by Bim activation and, interestingly, by NLRP3 inflammasome pathway activation [208].

Sesamin is a lignan shown to have anti-inflammatory and anti-neoplastic effects [209]. Meng et al. demonstrated that sesamin significantly inhibited the growth of EL4 lymphoma cells in a tumor-bearing mouse model by inducing apoptosis, pyroptosis (assessed by the relative protein expression levels of cleaved-caspase 1, NLRP3, and IL-1β p17), and autophagy. They showed that after the sesamin treatment of EL4 cells, autophagy preceded apoptosis and pyroptosis, while autophagy inhibition resulted in the downregulation of apoptosis and pyroptosis. Overall, these findings suggest that for murine T-cell lymphoma, sesamin exerts its antitumor effects via autophagy-mediated induction of apoptosis and pyroptosis [210].

## 9. Discussion

Inflammasomes are cardinal components of the innate immune response, as PRRs inducing pro-inflammatory cytokine release and pyroptosis [2,3,4,5], while accumulating data highlight their additional implication in adaptive immune responses [60,61]. The interplay between inflammation, autoimmunity, and cancer is a research field of utmost interest [1]. In this setting, the implication of the NLRP3 inflammasome in carcinogenesis has gained increasing scientific attention, whereas its activation acts both as a tumor-promoting and a tumor-protective factor [20]. On the one hand, it can trigger an effective immune response against cancer cells, while on the other hand, hyperinflammation driven by a dysregulated inflammasome activation could favor a tumor-permissive microenvironment [35]. Of note, the effects of NLRP3 inflammasome activation can vary depending on the cell type its components are expressed and assembled.

Regarding hematological malignancies, the role of NLRP3 inflammasome aberrations originally gained interest in the field of myeloid neoplasms. Specifically, BM mononuclear cells (BMMNCs) from patients diagnosed with myelodysplastic neoplasms (MDS) are characterized by upregulated mRNA levels of NLPR3 inflammasome-related genes (i.e., *CASP1*, *NLRP3*, *IL-18*, and *IL-1β*) compared to BMMNCs from healthy controls [211], while their PBMCs also display increased levels of *CASP1* transcripts [212]. Apart from its possible implication in the pathogenesis of myelodysplasia, pyroptosis also seems to drive the MDS phenotype, given that mRNA levels of the IL-1 family cytokines and caspase-1 are higher in lower-risk MDS compared to higher-risk MDS [211,212]. In the case of MDS, S100A9 and oxidized mtDNA were proposed as alarmins responsible for NLRP3 inflammasome activation [211,213]. Overall, studies exploring the role of NLRP3 inflammasome in the pathogenesis of AML report an upregulation in NLRP3-associated genes, namely *NLRP3*, *ASC*, *CASP1*, *IL-18*, and *IL-1β*, both in BMMNCs and PBMCs, though results for specific components present variations between studies [214,215]. The reported HMGB1 upregulation in AML patients and its concurrent positive correlation with NLRP3 mRNA expression highlight its role as a driver of NLRP3 activation in the setting of AML progression [216]. Interestingly, NLRP3 and IL-1β mRNA expression levels were shown to be decreased, while IL-18 mRNA expression levels were shown to be increased in chronic myeloid leukemia (CML) compared to controls [217]. In the case of CML *CARD8*, *IL-18*, and *IL-1β,* single-nucleotide polymorphisms (SNPs) were correlated with risk stratification and molecular response after treatment [217]. Zhou et al. demonstrated that BM cells from patients with Philadelphia chromosome-negative myeloproliferative neoplasms (MPNs) showed a significantly increased expression in NLRP3 inflammasome-related genes (*NLRP3*, *NF-κB1*, *CARD8*, *IL-1β*, and *IL-18*) and the increased expression was associated with the JAK2V617F mutation, white blood cell counts and splenomegaly. As for malignancies of lymphoid origin, CASP1 and NLRP3 showed significantly higher expression in glucocorticoid-resistant primary leukemia cells from patients diagnosed with acute lymphoblastic (ALL) compared to cells sensitive to glucocorticoids [218]. Furthermore, certain *CARD8* SNPs in adults and *IL-1β* SNPs in children were correlated to increased susceptibility for ALL development [219,220]. mRNA expression in NLRP3 and caspase-1 was significantly downregulated in patients with newly diagnosed MM, while NLRP3 mRNA levels negatively correlated with β2-microglobulin levels and BM plasma cell infiltration, features characterizing poor prognosis and more progressed disease. Intriguingly, only IL-1β protein levels positively correlated with NLRP3 and caspase-1 mRNA expression, while no correlation was shown for IL-18 protein levels, which were reported to be higher in MM patients [221]. Hofbauer et al. reported that β2-microglobulin accumulation in myeloma-associated macrophages leads to NLRP3 inflammasome activation, with the subsequent IL-18 production driving MM cells’ proliferation [222]. Finally, *CARD8* SNPs were correlated with MM susceptibility and disease stage [223]. Given the heterogeneity of the multiple lymphoma subtypes, in this review, we try to underscore the role of the NLRP3 inflammasome in lymphomagenesis.

Deciphering the mechanisms of NLRP3 activation during normal lymphopoiesis is crucial for understanding its potential effect on neoplastic transformation. Data from animal models show that NLRP3 inflammasome activation shapes the balance between myelopoiesis and B-cell lymphopoiesis during aging in favor of the former, a process driven by the increase in BM adipose tissue [74,75]. Analogous to B-cell lymphopoiesis, T-cell maturation in the thymus seems to be affected by age-related changes since thymic involution was shown to be driven by lipotoxic DAMPs inducing thymic caspase-1 activation via the NLPR3 inflammasome [82]. Furthermore, NLRP3 inflammasome downregulation leads to the prevalence of κ over λ B-cells in the BM and the aberrant over-representation of B-cell subsets with innate-like properties in the periphery [31]. Depending on the type of microbial stimuli, the NLRP3 inflammasome can also drive either the secretion of pro-inflammatory cytokines or the production of IgM antibodies by mature B-cells [80]. We can, therefore, conclude that the NLRP3 inflammasome is implicated in the regulation of lymphopoiesis both in the stages of early lymphocyte development and subsequent maturation, with an interesting role in the lymphopoiesis impairment observed with aging.

Lymphomas arise from lymphocytes at different stages of differentiation, depending on the lymphoma subtype. For most lymphomas, specifically DLBCL [83,113], MALT lymphoma [89,114], MCL [90], PEL [91], and ΝΚ/Τ-cell lymphoma [129], NLRP3 inflammasome activation seems to exert a pro-tumorigenic effect. Interestingly, for CLL/SLL [92] and Burkitt lymphoma [93], NLRP3 inflammasome activation acts as a tumor-suppressive factor, while for CTCLs, NLRP3 inflammasome activation was associated with both pro- and anti-tumorigenic effects [120,124]. These contradictory findings require further elucidation through the prism of the complicated crosstalk of various pathways and molecules that possibly regulate the NLRP3 inflammasome during lymphomagenesis. We should also consider the differences in the TME of each lymphoma subtype, given the effect of the TME on NLRP3 inflammasome activation and vice versa.

For DLBCL, which is the most studied lymphoma subtype regarding the NLRP3 inflammasome, it can be proposed that NLRP3 inflammasome activation is driven by two main mechanisms: (i) lymphoma cell-intrinsic features favoring an autonomous signal and (ii) bidirectional interactions between lymphoma cells and the TME. As for the first mechanism, several polymorphisms of NLRP3 inflammasome-related molecules were shown to confer increased susceptibility to DLBCL development and were also correlated with patients’ survival, prognosis, and treatment response [101,102]. We should notice, though, that the functional consequences of these polymorphisms were not explored. Likewise, the upregulation of PRGs in the setting of DLBCL [102] was not validated at the protein level. We can, therefore, only assume that these genetic aberrations predispose lymphoma cells to the initiation and perpetuation of NLRP3 inflammasome signaling via either structural differences or the quantitative increase in its related protein components. When it comes to the interplay of lymphoma cells and the TME, BAFF, mainly produced by monocytes and dendritic cells [224], was shown to drive NLRP3 inflammasome priming and activation both in normal and neoplastic B-cells via the BAFF/BAFF-R axis [113], possibly contributing to the proliferation advantage and apoptotic resistance of malignant B-cell clones. This upregulated NLRP3 inflammasome activation in lymphoma cells can act to modify the antitumor properties of the TME immune cells since it was shown to lead to PD-L1 upregulation in lymphoma cells and, therefore, impaired antitumor cytotoxic T-cell immunity, while it might also lead to an increased proportion of PD-1/TIM-3-expressing T-cells, MDSCs, TAMs, and regulatory T cells, overall favoring lymphoma progression [86].

Chronic antigenic stimulation and persistent inflammatory signaling can lead to the emergence of clonal B-cell populations, which may further transform into neoplastic B-cells. Such is the case of MALT lymphoma development in the setting of autoimmunity, where NLRP3 inflammasome activation was mainly investigated in the setting of SS, an autoimmune disease correlating with lymphomagenesis due to chronic antigenic stimulation [225]. An overactivated P2X7R-NLRP3 inflammasome axis was reported in SS patients who develop lymphoma [89], while persistent NLRP3 inflammasome activation in monocytes and macrophages driven by cf-DNA and nucleosomes arising from epithelial SG cells was also described [118]. Moreover, though immediate correlations between BAFF levels and NLRP3 inflammasome activation were not studied in the setting of SS-related lymphomagenesis, it can be proposed that the increased BAFF levels, documented in SS patients with increased risk for lymphoma development or concomitant lymphoma [114], may predispose to lymphoma progression via BAFF-induced NLRP3 inflammasome activation in B-cells [113], as described previously.

As opposed to B-cell lymphomas, where overall NLRP3 inflammasome activation seems to support lymphomagenesis for the majority of the studied lymphoma subtypes, studies for T-cell lymphomas reached diverse conclusions. Research mainly focused on the role of the NLRP3 inflammasome in the pathogenesis of CTCLs. CTCL patients demonstrate increased IL-18 plasma levels and increased *IL-18* mRNA levels in skin lesions, more likely induced after NLRP3 inflammasome activation, given the concomitant increase in caspase-1 mRNA in the affected skin. In this case, it was proposed that increased IL-18, in combination with the absence of IL-12, drives the characteristic Th2 skewing observed in CTCL [120]. Interestingly, other data support the inhibition of NLRP3 inflammasome assembly due to unassembled NLRP3 translocation to the nucleus, where it promotes *IL-4* transcription, which is associated with CTLCL lesion severity and progression [124]. As for Sézary syndrome, a rare leukemic type of CTCL, different expression patterns of NLRP3 and its effector cytokines were reported between different skin layers and LNs [127], a phenomenon that can be attributed to variable effects of NLRP3 inflammasome activation in the process of lymphomagenesis depending on the affected tissue.

The downstream effects of NLRP3 inflammasome activation are, in part, mediated by the secretion of its effector cytokines. Measuring these cytokines could mirror the degree of NLRP3 inflammasome activation. Several studies assessed IL-18 mRNA and protein expression in lymphoma patients either in serum or the affected tissues. For DLBCL, increased serum IL-18 was correlated with lower CR rates after treatment and decreased OS and DFS [83,84,85], while higher IL-18 tissue expression was characteristic of the more aggressive non-GCB DLBCL subtype [86]. In patients with SS-associated MALT lymphomas, increased *IL-18* mRNA levels in affected SGs were documented compared to SS patients without lymphoma, with higher IL-18 also correlating with the presence of GC-like structures [89]. Likewise, for CTCLs, elevated serum IL-18 and *IL-18* mRNA in skin lesions were reported [120], while for the rare NK/T-cell lymphomas, high IL-18 serum levels were associated with advanced-stage disease, the presence of hemophagocytosis, and poor treatment outcomes [128]. Of note for Sézary syndrome, there are discrepancies in the expression of IL-18 in serum and tissues or even in different parts of the same tissue [127]. Overall, these findings support that an overactivated NLRP3 inflammasome pathway acting via IL-18 promotes lymphomagenesis and confers a dismal prognosis. This is in line with the general notion that the NLRP3 inflammasome and its effector cytokines favor an immunosuppressive TME [226]. We should note, though, that IL-18 maturation can also be mediated by other mechanisms, such as cleavage by caspase-8 induced by FAS-ligand signaling [227], by chymases secreted by mast cells [228], or by granzyme B secreted by CD8+ T-cells [229]. Therefore, studies confirming the NLRP3-driven increase of IL-18 are necessary to prove a solid causal relationship.

Concerning lymphomagenesis, it is noteworthy that, among inflammasomes, research focused on NLRP3. This might be expected, considering that the NLRP3 inflammasome is the prototypical and best-characterized inflammasome. Nevertheless, in the case of lymphomagenesis, we should also ruminate on the different mechanisms that drive the activation of the different inflammasome subtypes. NF-κΒ, a known contributor to the pathogenesis of various lymphoma subtypes [230], is also a factor inducing NLRP3 inflammasome priming [3,4]. This can be one of the main reasons NLRP3 inflammasome is found to be dysregulated during lymphoma development and progression. Moreover, the role of mtROS in NLRP3 inflammasome priming [4,50,51,52] and the metabolic aberrations described in various lymphomas [231] might be additional contributors to the NLPR3 inflammasome dysregulation. The NLRP1 inflammasome is activated by double-stranded RNA, ultraviolet B, and ribotoxic stress response [232,233], while NLRC4 is induced by PAMPs from intracellular bacteria [41]. NLRC4 can directly interact with pro-caspase-1 through CAR-CARD interactions, leading to NLRC4-dependent cell death without efficient cytokine production, while ASC is required for caspase-1 activation and the cleavage of pro-IL-1β and pro-IL-18 [234]. AΙΜ2 does not belong to the NLR family of inflammasomes, though it contains a pyrin domain that enables ASC recruitment, while its activation by the detection of either host or microbial double-stranded DNA leads to the formation of a multi-protein caspase containing complex that drives PANoptosis instead of pyroptosis [42,235]. Common pathways concurrently implicated in lymphomagenesis and the activation of these inflammasomes have not been elucidated thus far. We should note, though, that there are few reports regarding AIM2 and NLRP1 [127,236].

Regarding therapeutics, despite the progress made, particularly in the last decade, in the treatment approaches for B-cell NHLs, there are still unmet needs, especially in the setting of relapsed/refractory disease, while for T-cell lymphomas, the available therapeutic modalities are still limited with a subsequent dismal prognosis for patients. Unraveling new pathways that contribute to lymphomagenesis offers the possibility for further therapeutic targeting. Exploiting the role of the NLRP3 inflammasome in the pathogenesis of lymphoma is an attractive field for the development of new treatment strategies, though research results are still immature. So far, no drugs that specifically target the NLRP3 inflammasome have been developed. Nevertheless, drugs predominantly tested for their potential to inhibit other pathways implicated in lymphoma development were shown to exert part of their action via interference with the NLRP3 inflammasome pathway. The multikinase inhibitor Luxeptinib, apart from its main function to inhibit BCR activation pathways, was also shown to impair the ability of the NLRP3 inflammasome to cleave and activate caspase-1 [203,204]. This can serve as an example of combining the blockade of both BCR and inflammatory signaling to achieve better results in B-cell NHL. Of course, the efficacy of this strategy in the clinical setting remains to be proven, given that a phase 1a/b trial of Luxeptinib in patients with relapsed/refractory B-cell malignancies is still ongoing [201]. Another interesting approach could be the study of drugs that have a concomitant effect both on NLRP3 inflammasome activation and the expression of molecules, such as PD-1 and PD-L1, to enhance the effect of immunotherapy. HPK1 inhibitors are such an example since, apart from downregulating NLRP3, they can also induce PD-1 and PD-L1 expression, making their combination with anti-PD-1/PD-L1 immunotherapy attractive [206]. We should consider, though, that these results derive from in vitro studies and need further validation. As opposed to lymphomas of B-cell origin, in the case of T-cell lymphomas, enhancing the activation of the NLRP3 inflammasome is what exerts antitumor effects. Specifically, drugs that were tested in humans (i.e., HDACis) and murine models (i.e., sesamin) were shown to induce NLRP3-mediated cell death [208,210].

Though the investigation of NLRP3 inflammasome therapeutic targeting in the setting of lymphoma is still at a premature stage, its activity could be modulated by multiple approaches: (a) the abrogation of upstream signaling pathways, (b) the direct inhibition of its components, and (c) the antagonism of its end-products. As for the first approach, it could further be divided into targeting the NLRP3 inflammasome priming step or activation step. IL-1 receptor-associated kinase 4 (IRAK-4) is a key mediator of the TLR signaling processes [237], while TLR4 signaling induces the NLRP3 inflammasome priming step via NF-κB. Emavusertib, a selective IRAK-4 inhibitor, is under clinical investigation to overcome the persistent survival signaling through IRAK-4 in B-cell lymphomas [238]. Multiple therapeutic modalities were explored in terms of NF-κΒ inhibition for lymphoid neoplasms by directly targeting NF-κB components or indirectly by inhibiting upstream signaling pathways’ components (reviewed in [239]). Theoretically, all these approaches could affect NLRP3 inflammasome priming, though specific experimental investigations are not available. As for the step of NLRP3 inflammasome activation, ion efflux, ROS, or oxidized mtDNA generation, and lysosomal destabilization/cathepsin B constitute possible therapeutic targets. For example, glyburide, a compound investigated in the setting of gestational diabetes mellitus [240], or P2RX7 inhibitors, which were shown to exert an antitumor effect in murine melanoma models [241], are known to inhibit potassium efflux. VI-16, a synthetic flavonoid derivative, reduces mtROS, thus inhibiting NLRP3 activation through TXNIP, presenting another attractive therapeutic modality to be investigated [242]. Type I interferons inhibit NLRP3 activation through multiple mechanisms: increased nitric oxide production, IL-10 transcription, and other types of STAT1-mediated transcription [243]. IFN-α based immunotherapies were used for the treatment of lymphomas both of B- and T-cell origin [244,245], possibly exerting a part of their therapeutic effect via mitigating the degree NLRP3 inflammasome activation. Research demonstrated that BTK activity is important for NLRP3 inflammasome activation [246]. It can, therefore, be proposed that the widely used for B-cell lymphoma treatment BTK inhibitors, apart from other well-studies pathways [247], exert their action via their contribution to the inhibition of the NLRP3 inflammasome activation. Regarding direct NLRP3 inflammasome inhibition, certain drugs, namely MCC950, CY09, OLT1177, INF39, NT-0167, oridonin, tranilast, inzomenil, somalix were developed and studied either at the preclinical level or in phase I/II clinical trials mainly in the setting of inflammatory disease with encouraging results [248]. Their efficacy in the setting of neoplastic diseases, such as lymphoma, is yet to be explored, though it presents an attractive perspective. BOT-4-one, a benzoxantiole derivative, was shown to inhibit NLRP3 inflammasome activation via NLRP3 inflammasome alkylation and subsequent impaired ATPase activity, leading to the obstruction of its assembly [249]. Kim et al. reported that BOT-4-one suppressed L540 lymphoma cell survival and proliferation via the inhibition of JAK3/STAT3 signaling [250]. Although, in this study, the effects of BOT-4-one on NLRP3 inflammasome assembly were not investigated, the observed antitumor effect on lymphoma cells could partly be attributed to the compound’s inhibitory properties on NLRP3 inflammasome assembly, rendering it a possible candidate for future clinical investigation. Considering the role of TRIM family proteins in regulating NLPR3 inflammasome at multiple levels [134], further therapeutic prospectives arise. In the setting of clinical studies, cytokine blockade seems to be the most effective approach, with research focusing on IL-1β mediated signaling as a target. The IL-1R antagonist, Anakinra, is under investigation in clinical trials including patients mainly with solid tumors (reviewed in [173]). As for hematologic malignancies, a phase II clinical trial of Anakinra in patients with smoldering or indolent MM at risk of progression documented encouraging results in terms of PFS [251]. Given the B-cell-derived origin of plasma cells as well as the pathways shared in the pathogenesis of lymphoid and plasma cell malignancies, targeting IL-1β signaling presents a potential therapeutic approach for B-cell lymphomas. To summarize, research investigating the role of the NLRP3 inflammasome in lymphomagenesis is still limited as compared to other types of malignancies. NLRP3 inflammasome activation seems to exert a dual role, pro- and antitumorigenic, depending on the type of lymphoma. The exact mechanisms that regulate NLRP3 inflammasome activation, as well as the exact downstream effects of either its up- or downregulation in the neoplastic lymphocytes and their microenvironment, need further elucidation. This will not only expand our knowledge regarding the intricate lymphoma pathogenesis but will also contribute to the development of new therapeutic strategies.

## 10. Conclusions

Besides the extensively studied role of the NLRP3 inflammasome in innate immunity, increasing evidence highlights its implication in various malignancies, amongst them lymphomas. Understanding the mechanisms that regulate the NLRP3 inflammasome during normal lymphopoiesis is crucial for deciphering its entanglement in lymphomagenesis. For most lymphoma subtypes, NLRP3 inflammasome activation was shown to act as a tumor-promoting factor, but a tumor-suppressive role was also described. Aberrations of the NLRP3 inflammasome pathway not only affect lymphoma cell growth and proliferation but also modulate the TME. Further research is imperative to define the regulators of NLRP3 inflammasome activation in lymphoma, while preliminary data support that its targeting could provide additional treatment modalities to cover unmet therapeutic needs.

## Figures and Tables

**Figure 1 ijms-25-02369-f001:**
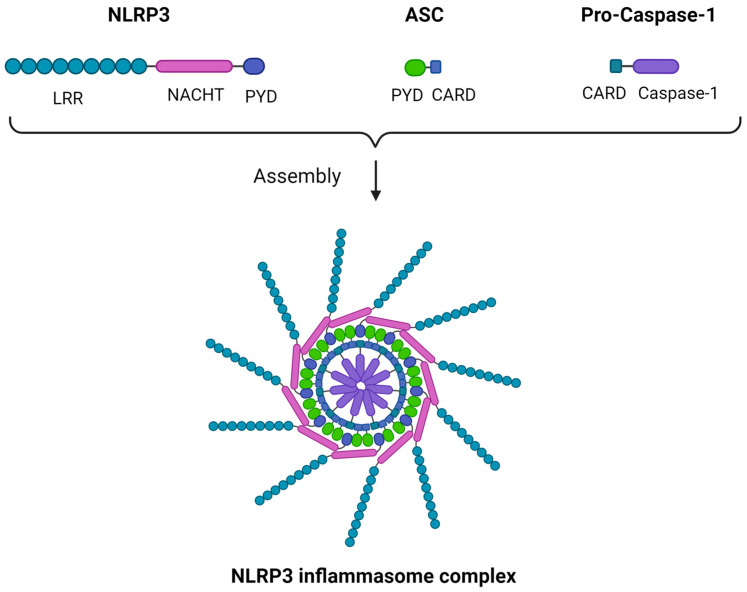
The NLRP3 inflammasome structure. Three main components comprise the NLRP3 inflammasome: (i) NLRP3, which acts as a sensor and comprises an LRR domain, a NACHT domain, and a PYD; (ii) ASC, which acts as an adaptor and comprises a PYD and a CARD; (iii) pro-caspase-1, which comprises a CARD and caspase-1. The ASC adaptor forms a speck-like structure that binds the NLRP3 sensor to the pro-caspase-1 component through PYD-PYD and CARD-CARD interactions. Multiple ASC dimers can further polymerize, forming a supramolecular assembly, the NLRP3 inflammasome complex. ASC, apoptosis-associated speck-like protein containing a CARD; CARD, caspase activation and recruitment domain; LRR, leucine-rich repeats; NACHT, NAIP (neuronal apoptosis inhibitor protein), C2TA (MHC class 2 transcription activator), HET-E (incompatibility locus protein from Podospora anserina), TP1 (telomerase-associated protein); NLRP3, Nod-like receptor family pyrin domain containing 3; PYD, pyrin domain. Created with BioRender.com. https://app.biorender.com/ (accessed on 2 January 2024).

**Figure 2 ijms-25-02369-f002:**
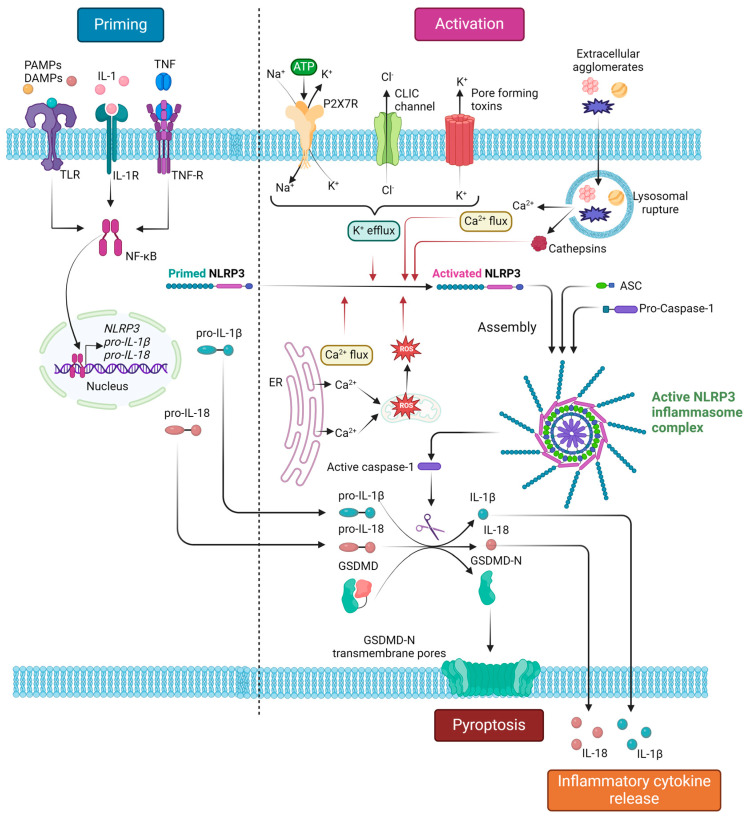
NLRP3 inflammasome complex canonical activation and downstream responses. The activation process of the NLRP3 inflammasome consists of two main signals: (i) the priming signal, which is mainly induced by PAMP or DAMP recognition by TLRs and leads to NF-κΒ-mediated transcriptional upregulation of *NLRP3*, *pro-IL-1β* and *pro-IL-18*; (ii) the activation signal, which is induced by various stimuli, including extracellular ATP, pore-forming toxins and extracellular agglomerates (monosodium urate crystals, cholesterol crystals, asbestos, silica), that lead to molecular or cellular events, namely K^+^ efflux, Ca^2+^ flux, ROS generation, and lysosomal damage, known to activate the NLRP3 inflammasome. Inflammasome activation leads to auto-cleavage and formation of the active caspase-1, which, in turn, proteolytically cleaves its substrates pro-IL-1β, IL-18, and GSDMD. The cleaved bioactive IL-1β and IL-18 are subsequently released extracellularly, resulting in an inflammatory response, while the cleaved GSDMD-N polymerizes to form transmembrane pores, leading to pyroptosis. ASC, apoptosis-associated speck-like protein containing a CARD (caspase activation and recruitment domain); CLIC, chloride intracellular channel; DAMPs, damage-associated molecular patterns; IL, interleukin; GSDMD, gasdermin D; GSDMD-N, gasdermin D N-terminal; NF-κΒ; nuclear factor kappa-light-chain-enhancer of activated B-cells; NLRP3, Nod-like receptor family pyrin domain containing 3; PAMPs, pathogen-associated molecular patterns; TLR, toll-like receptor; TNF, tumor necrosis factor; TNF-R, TNF receptor. Created with BioRender.com. “https://app.biorender.com/ (accessed on 2 January 2024)”.

**Figure 3 ijms-25-02369-f003:**
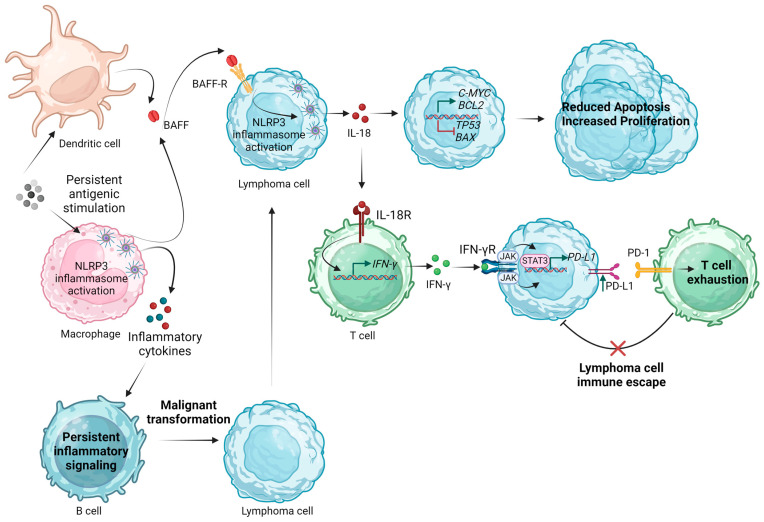
Proposed mechanisms of NLRP3 inflammasome implication in the pathogenesis and progression of B-cell non-Hodgkin lymphomas. BAFF, B-cell activating factor; IFN-γ, interferon γ; JAK, Janus kinase; IL, interleukin; NLRP3, Nod-like receptor family pyrin domain containing 3; PD-1, programmed cell death protein; PD-L1, programmed death-ligand 1; R, receptor; STAT3, signal transducer and activator of transcription 3. Created with BioRender.com. “https://app.biorender.com/ (accessed on 2 January 2024)”.

**Figure 4 ijms-25-02369-f004:**
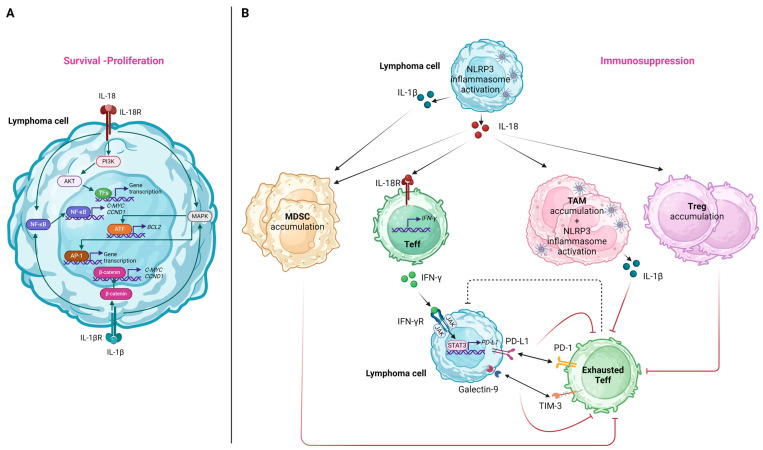
The effects of NLRP3 inflammasome activation and its effector cytokines on lymphoma cells (**A**) and immune cells of the tumor microenvironment (**B**). (**A**) IL-18 and IL-1β released after NLRP3 inflammasome activation interact with their receptors on lymphoma cells, inducing signaling pathways that promote the expression of genes related to apoptosis inhibition, survival, and proliferation. (**B**) The pro-inflammatory cytokines secreted after NLRP3 inflammasome activation both in lymphoma cells and immune cells favor the accumulation of immunosuppressive immune cell subpopulations and promote effector T-cell exhaustion, contributing overall to immune evasion of the lymphoma cells. AP-1, activator protein 1; ATF, activating transcription factor-2; CCND1, cyclin D1; IFN, interferon; IL, interleukin; MDSC, myeloid-derived suppressor cell; NF-κB, nuclear factor kappa-light-chain-enhancer of activated B-cells; PI3K, phosphoinositide 3-kinase; PD-L1, programmed death-ligand 1; PD-1, programmed cell death protein 1; STAT3, signal transducer and activator of transcription; TAM, tumor-associated macrophage; Teff, T effector; TIM-3, T-cell immunoglobulin and mucin-domain containing 3; Treg, T regulatory; TFs, transcription factors. Created with BioRender.com. “https://app.biorender.com/ (accessed on 2 January 2024)”.

**Table 1 ijms-25-02369-t001:** NLRP3 inflammasome in the pathogenesis of lymphomas.

Lymphoma Subtype	Type of Sample Studied	Effect of NLRP3 Inflammasome or Related Molecules	Studied NLRP3 Inflammasome Components or Related Molecules	Proposed Mechanism	Findings	Possible Clinical Relevance	Ref.
DLBCL	Pfeiffer cell line	Tumor-promoting	IL-18	NLRP3 inflammasome activation induces: *C-MYC* and *BCL-2* upregulation *TP53* and *BAX* downregulation	NLRP3 inflammasome activation inhibited dexamethasone-induced apoptosis via shifting the balance of *BCL-2*/*BAX* expression	Resistance to dexamethasone	[83]
DLBCL	73 patients: serum samples	Tumor-promoting	IL-18	IL-18-mediated pro-tumorigenic effect	Higher mean IL-18 levels correlated with lower CR (*p* = 0.465) and 3-year DFS rates (*p* = 0.127) after R-CHOP Lower mean IL-18 levels correlated with more favorable OS rates (*p* = 0.008)	Prognostic value for survival and treatment response	[85]
DLBCL	35 patients and 35 healthy controls: lymphoid tissue samples	Tumor-promoting	IL-18, PD-L1	IL-18-mediated pro-tumorigenic effect IL-18-mediated immune exhaustion	Higher IL-18 expression in DLBCL tissues compared to normal lymphoid tissues Higher IL-18 expression in DLBCLs with non-GCB phenotype compared to those with GCB phenotype Higher PD-L1 expression in DLBCL tissues compared to normal lymphoid tissues	Association with more aggressive disease Association with immune exhaustion—possible application for ICB therapy	[86]
DLBCL	Cell lines, murine model	Tumor-promoting	NLRP3, PD-L1, pSTAT3	NLRP3 induced IL-18 secretion drives PD-L1 upregulation via pSTAT3	NLRP3 inflammasome activation led to PD-L1 upregulation and T-cell decrease NLRP3 inflammasome blockade suppressed lymphoma growth and ameliorated antitumor immunity by downregulating PD-L1 in the TME and decreasing the proportion of immunosuppressive immune cells pSTAT3 protein levels decreased after IL-18 and NLPR3 inflammasome inhibition	Association with immune exhaustion—possible application for ICB therapy	[86]
DLBCL	GEO and TCGA gene expression datasets for DLBCL and normal B cells	Tumor-promoting	PRGs		Upregulated: *PYCARD*, *IRF1*, *GZMA*, *GSDMD*, *GSDMC*, *GPX4*, *CHMP2A*, *CASP5*, *CASP1, IL-18* Amplified CNV status: *BAK1* Three distinct clusters identified and associated with prognosis	Risk stratification	[102]
DLBCL	TCGA dataset: dataset including 48 tissue samples of DLBCL patient	Tumor-promoting	Multiple inflammasome components	NLRP3 inflammasome upregulation in macrophages drives a proinflammatory TME that favors lymphomagenesis	Significant upregulation in *AIM2*, *ALK*, *IRF3*, *4* and *8*, *NFKB1* and *2*, *NOD2*, *NLRP1* and *3*, *CASP1* and *5*, *CARD8* and *9* in M0 and M1 macrophages		[100]
Aggressive B-cell NHL	46 patients and 20 healthy controls: serum samples	Inconclusive	IL-18		No significant difference in IL-18 serum levels between patients and controls (*p* = 0.261) Νο difference in IL-18 serum levels before and after chemotherapy Higher IL-18 serum levels were higher in patients with higher LDH compared to those with normal LDH (*p* = 0.045)	Possible association with high-risk lymphoma features	[88]
B-cell NHL	281 patients with B-cell NHL (BM samples) and 385 healthy controls (PB samples)	Tumor-promoting	IL-1β, IL-18, NF-κΒ, CARD8	NF-κΒ mediated NLRP3 inflammasome activation IL-18-mediated pro-tumorigenic effect	IL-18 (rs1946518) and NF-κB-94 ins/del (rs28362491) gene polymorphisms contributed to increased risk for B-cell NHL (*p* < 0.0001 and *p* = 0.0029, respectively) The AA genotype of the CARD8 rs2043211 polymorphism correlated with poorer survival (*p* = 0.0381) Τhe TT genotype of CARD8 (rs2043211) was observed in patients with higher LDH levels, clinical stages III-IV, and IPI scores 3–5	Biomarkers for lymphoma susceptibility Possible association with high-risk lymphoma features	[101]
B-cell NHL	Primary B cells and lymphoma cell lines	Not investigated Possibly tumor-promoting	NLRP3, caspase-1, IL-1β, cIAP1-TRAF2, Src	BAFF/BAFF-R axis induces cIAP1-TRAF2-mediated NLRP3 priming signals and Src activity-dependent ROS mediated NLRP3 inflammasome activation signals	Treatment with BAFF led to an increase in NLRP3 mRNA expression but also in increased NLRP3 inflammasome activation (as proven by the increase in active caspase-1 and IL-1β levels)		[113]
NHL	27 patients: serum samples	Tumor-promoting	IL-18	IL-18-mediated pro-tumorigenic effect	Patients with IL-18 serum levels >2000 pg/mL compared to those with IL-18 serum levels <1000 pg/mL showed lower CR rates (33.3% versus 85.7%) and lower median OS (3.5 months versus 45.5 months) after CHOP chemotherapy	Prognostic value for survival and treatment response	[84]
MALT lymphoma	45 SS patients and 25 sicca controls: salivary gland tissue specimens	Tumor-promoting	P2X7R, NLRP3, caspase-1, IL-18, IL-1β	P2X7R-mediated NLRP3 inflammasome activation IL-18-mediated pro-tumorigenic effect	Higher mRNA expression for P2X7R, NLRP3, caspase-1, IL-18, and IL-1β in patients with SS who developed MALT-NHL over the follow-up More pronounced P2X7R protein expression in SS patients developing MALT-NHL Three-fold higher glandular expression of IL-18 in MALT-NHL than in controls or the other patients with SS.	Biomarker for lymphoma development in SS	[89]
MALT lymphoma	76 SS patients, 11 non-SS disease controls, and 30 healthy controls: salivary gland tissue specimens and serum samples	Tumor-promoting	NLRP3, caspase-1, IL-18, IL-1β, ASC	Cell-free and extranuclear DNA-mediated NLRP3 inflammasome activation IL-18-mediated pro-tumorigenic effect	Systemic activation of the NLRP3 inflammasome and significantly increased serum IL-18 and ASC levels in SS patients at high risk for lymphoma development and those with established lymphoma	Biomarker for lymphoma development in SS	[118]
MCL	7 lymphoid tissue samples from patients	Tumor-promoting	IL-18	IL-18-mediated pro-tumorigenic effect	*IL-18* gene upregulation	Further MCL subtyping	[90]
PEL	BJAB cells, BCBL-1 PEL cells	Not investigated Possibly tumor-promoting	Caspase-1, IL-1β, IL-18		Activation of caspase-1 and cleavage of pro-IL-1β and pro-IL-18 detected in lymphoma cells induced by KSHV		[91]
CLL/SLL	23 patients: B cells from PB samples	Tumor-suppressive	P2X7R, ASC, NLRP3	P2X7R-mediated NLRP3 inflammasome activation	NLRP3 overexpression correlated with inhibition of cell proliferation and induction of apoptosis NLRP3 downregulation contributed to lymphomagenesis	Biomarker for disease aggressiveness or progression	[92]
Burkitt lymphoma	HH514-16 and CLIX-FZ cell lines	Tumor-suppressive	NLRP3, HMGB1	HMGB1 protein sustains ZEBRA expression via the NLRP3 inflammasome	EBV lytic phase Burkitt lymphoma cells express high levels of HMGB1 HMGB1 regulates NLRP3 inflammasome-mediated ZEBRA expression		[93]
CTCL	95 patients: plasma samples 20 patients: skin tissues	Tumor-promoting	IL-18, caspase-1	Increased IL-18 level can favor skewing of CD4+ T cells to the characteristic Th2 phenotype	Increased IL-18 and caspase-1 plasma levels in patients compared to healthy controls Increased IL-18 and caspase-1 mRNA levels in skin lesions from patients compared to healthy skin	Biomarker	[120]
CTCL	53 patients, 12 healthy controls, 10 patients with psoriasis, 12 patients with atopic dermatitis: skin tissues	Tumor-promoting	NLRP3, IL-4	NLRP3 translocation to the nucleus of the malignant CD4+ T cells, where it binds to the human IL-4 promoter in duces IL-4 production, which promotes the characteristic Th2 phenotype	IL-4 production mediated by NLRP3 increased with lesion severity and associated with disease progression	Association with disease severity and progression	[124]
Sézary syndrome	28 patients, 19 erythroderma patients, and 40 healthy donors: skin tissues, PB samples, serum samples, lymphoid tissue samples	Tumor-promoting (dependent on the affected tissue)	NLRP3, IL-1β, IL-18, AIM2, NLRP1	IL-18-mediated pro-tumorigenic effect—though its production might not be exclusively mediated by NLRP3 inflammasome	Increased IL-1β and low IL-18 levels in the epidermal skin layer of patients Increased IL-18 expression and no difference in IL-1β in the dermal skin layers of patients compared to controls Equal NLRP3 and AIM2 expression in the skin among the different groups Increased NLRP1 expression in the skin of patients Increased IL-18 serum levels in patients Upregulation in IL-18 and downregulation in IL-1β in the LNs of patients with advanced-stage disease	Diagnostic biomarker Association with disease severity	[127]
NK/T-cell lymphoma	114 patients: serum samples	Tumor-promoting	IL-18	IL-18-mediated pro-tumorigenic effect	High IL-18 serum levels associated with stage III/IV disease, presence of hemophagocytosis, and poor treatment outcomes. OS and PFS were significantly lower for the high IL-18 group compared to the low IL-18 groups (*p* < 0.001) High IL-18 serum levels were independently prognostic for survival in multivariate analysis	Biomarker of hemophagocytosis Prognostic biomarker	[128]
NK/T-cell lymphoma	3 NK/T-cell lymphomas, 7 infectious mononucleosis cases, 6 chronic active EBV infection cases: lymphoid tissue miRNA expression datasets	Not investigated	hsa-miR-372-5p	Hsa-miR-372-5p may target NIMA-related kinase 7 to regulate NLRP3 inflammasome activation	Hsa-miR-372-5p regulates the NIMA-related kinase 7 and IL-1β release	Biomarker for EBV-associated disease	[129]
Various lymphoma subtypes (B-cell NHL, T-cell lymphoma, HL)	Lymphoma tissues: 68 patients (46 newly diagnosed, 22 treated) and 40 controls Plasma samples: 35 lymphoma patients and 15 controls	Tumor-promoting	IL-18	IL-18-mediated pro-tumorigenic effect	Higher IL-18 mRNA (*p* = 0.0288) and protein levels (*p* < 0.0001) in tissues of newly diagnosed lymphoma patients compared to controls Decrease in IL-18 mRNA (*p* = 0.0366) and protein levels (*p* = 0.0098) in tissues of patients with remission after chemotherapy Elevated plasma IL-18 protein levels in newly diagnosed lymphoma patients compared to controls Decreased plasma IL-18 protein levels after chemotherapy remission (*p* = 0.0098)	Diagnostic and prognostic biomarker	[83]

AIM2, absent in melanoma 2; ASC, apoptosis-associated speck-like protein containing a CARD; BM, bone marrow; CARD, caspase activation and recruitment domain; CHOP, cyclophosphamide, hydroxydaunorubicin, oncovin, prednisone; cIAP1, cellular inhibitor of apoptosis protein 1; CLL, chronic lymphocytic leukemia; CNV, copy number variations; CR, complete remission; CTCL, cutaneous T cell lymphoma; DFS, disease-free survival; DLBCL, diffuse large B-cell lymphoma; EBV, Epstein–Barr virus; GCB, germinal center B cell-like; GEO, gene expression omnibus; HL, Hodgkin lymphoma; HMGB1, high mobility group box 1; ICB, immune checkpoint blockade; IL, interleukin; IPI, international prognostic index; KSHV, Kaposi’s sarcoma-associated herpesvirus; LDH, lactate dehydrogenase; MALT, mucosa-associated lymphoid tissue; MCL, mantle cell lymphoma; NF-κB, nuclear factor kappa-light-chain-enhancer of activated B-cells; NHL, non-Hodgkin lymphoma; NIMA, never-in-mitosis gene A; NK, natural killer; NLRP, Nod-like receptor family pyrin domain containing; OS, overall survival; PB, peripheral blood; PD-L1, programmed death-ligand; PEL, primary effusion lymphoma; PRGs, pyroptosis-related genes; pSTAT3, phosphorylated signal transducer and activator of transcription 3; P2X7R, P2X purinoceptor 7; R, rituximab; SLL, small lymphocytic lymphoma; SS, Sjögren syndrome; TCGA, The Cancer Genome Atlas; Th2, T helper 2; TME, tumor microenvironment; TRAF2, TNF receptor-associated factor 2. Created with BioRender.com. “https://app.biorender.com/ (accessed on 2 January 2024)”.

## Data Availability

Data Availability Statement: Data are contained within the article.

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
