# Peer review of "Unraveling the Role of the NLRP3 Inflammasome in Lymphoma: Implications in Pathogenesis and Therapeutic Strategies"

_ijms, 2024, doi:10.3390/ijms25042369_

Round 1
Reviewer 1 Report
Comments and Suggestions for Authors
This review is well-organized and well written. All topics are covered satisfactorily.
Author Response
We sincerely appreciate your valuable feedback and thank you for your positive comments.
Reviewer 2 Report
Comments and Suggestions for Authors
Upon reviewing the article titled "Unraveling the Role of the NLRP3 Inflammasome in Lymphoma: Implications in Pathogenesis and Therapeutic Strategies," a number of suggestions and recommendations are possible. This article delves deeply into the function of NLRP3 inflammasome in lymphoma, emphasizing its pathophysiological significance and possible treatment approaches.
Observations:
This essay deftly presents inflammasomes as essential elements of the innate immune system, with NLRP3 being the most thoroughly researched.
1. Inflammasomes in Immune Responses: The article introduces inflammasomes as crucial components of the innate immune system, with NLRP3 being the most well-studied. This setup is vital for understanding the role of NLRP3 inflammasome in lymphoma.
2. Structure and Activation: This study describes the physiological and pathological processes in which the inflammasome is involved, such as cancer and autoimmunity. This review also highlights the mechanisms underlying inflammasome activation.
3. The function of the NLRP3 inflammasome in lymphocyte development, specifically B-cell lymphopoiesis, is discussed.
Lymphocyte Development, **. In secondary lymphoid tissues, it addresses how early B-cell development and differentiation are affected by NLRP3 inflammasome activation.
4. This research offers significant insights into the role of NLRP3 inflammasome activation in the development of B cell non-Hodgkin lymphoma (NHL) and how it affects the lymphoma microenvironment. In the tumor microenvironment, it addresses function of NLRP3 in drug resistance, tumor growth, and immune interactions.
5. Genetic and Environmental variables NLRP3 inflammasome activation in B-cell NHL is driven by genetic changes and environmental variables, which have been the subject of significant investigation. It investigates how inflammatory signaling, genetic polymorphisms, aberrations in B-cell growth factors, and B-cell NHL development are related.
Recommendations:
1. While the article focuses on B-cell NHL, expanding the discussion to include other hematological malignancies might provide a more comprehensive understanding of the role of the NLRP3 inflammasome in cancer.
2. The majority of the research available is preclinical or is based on an animal model. These arguments could be strengthened and given more direct application to human lymphoma by including more results from clinical trials and human studies.
3. Although the role of NLRP3 in lymphomagenesis is discussed in the paper, a more thorough investigation of the molecular pathways by which NLRP3 affects immune system and lymphoma cells may prove advantageous.
4. The article addresses the therapeutic targeting of the NLRP3 inflammasome; nevertheless, additional comprehensive discussions on existing and prospective therapeutic strategies, including particular medications and treatment modalities, would augment its pragmatic utility.
5. Analyzing NLRP3 in lymphoma in contrast to other inflammasomes may help to clarify some of its distinct features.
The paramount importance of considering the long-term effects and safety of targeting the NLRP3 inflammasome in therapeutic endeavors cannot be emphasized enough., particularly considering the role that inflammasomes play in normal immunological activities.
Overall, this study offers a comprehensive and perceptive analysis of the function of NLRP3 inflammasome in lymphoma, providing important details for understanding the etiology of the disease and possible targets for treatment. Its value in the field might be further increased by broadening its scope to encompass a greater number of studies, providing more in-depth mechanistic insights, and emphasizing therapeutic implications.
Comments on the Quality of English LanguageThe authors have a done a satisfactory work and may need some addition to there study and working out on comments.
Author Response
We would like to express our sincere gratitude for your diligent review of our manuscript. Your insightful comments and suggestions have been invaluable in refining the quality of our work. Below, we address each of your comments in a point-by-point manner.
Recommendation 1: “While the article focuses on B-cell NHL, expanding the discussion to include other hematological malignancies might provide a more comprehensive understanding of the role of the NLRP3 inflammasome in cancer.”
Response to Recommendation 1
Thank you for sharing your perspective. Though describing the role of NLRP3 inflammasome in the pathogenesis of other hematological malignancies was beyond the scope of our review, we have included a relevant part in the discussion section (lines 886-926). We hope that this will enhance comprehension of its multiple implications in the development of hematopoietic neoplasms irrespective of cell origin.
Recommendation 2: “The majority of the research available is preclinical or is based on an animal model. These arguments could be strengthened and given more direct application to human lymphoma by including more results from clinical trials and human studies.”
Response to Recommendation 2
We sincerely appreciate your recommendation. The investigation of NLRP3 inflammasome implication in the process of lymphomagenesis is an evolving field. In the setting of its implication in the pathogenesis of lymphoma, the majority of the published studies included human samples. To make that more evident we have included a table (Table 1) that summarizes all the relevant research findings pointing out the nature of the samples used. We also present in the table a column referring to the clinical significance of these findings. By reviewing the literature once again after your recommendation, we were able to detect only one study regarding NLRP3 inflammasome implication in DLBCL development, which applied bioinformatics analysis in a TCGA dataset including 48 tissue samples of DLBCL patients, that we have not included in the initial manuscript (L. Serna et a l. Front Immunol. 2023 May 2:14:1048567. doi: 10.3389/fimmu.2023) We have therefore incorporated it in the revised manuscript version. As for treatment approaches, research is still premature, justifying the paucity of available data from clinical trials. Nevertheless, we have included in the discussion section a part (lines 11078-1129) exploring therapeutic modalities targeting the NLRP3 inflammasome-related components, regulatory pathways, or downstream effectors in the setting of clinical trials for inflammatory diseases or other neoplasms, that could also be investigated in lymphomas, while we also review therapeutic agents that predominantly target other pathways in lymphoma therapeutics but may also be implicated in NLRP3 inflammasome regulation.
Recommendation 3: “Although the role of NLRP3 in lymphomagenesis is discussed in the paper, a more thorough investigation of the molecular pathways by which NLRP3 affects the immune system and lymphoma cells may prove advantageous.”
Response to Recommendation 3
We would like to thank you for your attention to detail and your valuable suggestion. Following your recommendation, we have included a new section in the revised manuscript (section 6) entitled “Molecular pathways by which NLRP3 inflammasome affects lymphoma cells and the immune TME”, as well as a relevant figure (Figure 4).
Recommendation 4: “The article addresses the therapeutic targeting of the NLRP3 inflammasome; nevertheless, additional comprehensive discussions on existing and prospective therapeutic strategies, including particular medications and treatment modalities, would augment its pragmatic utility.”
Response to Recommendation 4
Thank you for your insightful recommendation. A relevant part (lines 1078-1129) has been included in the discussion section to cover the issue of prospective therapeutic modalities, based on strategies targeting different parts of the NLRP3 inflammasome activation pathway, as well as with the discussion of data concerning treatment approaches on inflammatory or other neoplastic diseases that could also be investigated in the setting of lymphoma.
Recommendation 5: “Analyzing NLRP3 in lymphoma in contrast to other inflammasomes may help to clarify some of its distinct features.”
Response to Recommendation 5
We appreciate your suggestion. A relevant part addressing it is included in the discussion (lines 1028-1049).
Reviewer 3 Report
Comments and Suggestions for Authors
The manuscript is exceptionally well-crafted, and the illustrations are truly commendable. I have a single suggestion to enhance its impact further. As reported, TRIM family proteins regulate the NLRP3 inflammasome pathway and are implicated in various blood cancers. I recommend including a dedicated section on the crosstalk of TRIM family proteins with NLRP3 in lymphoma. This addition would undoubtedly elevate the manuscript, providing a more comprehensive perspective and potentially uncovering novel insights into the intricate connections within the NLRP3 inflammasome pathway in the context of lymphoma.
Author Response
We appreciate the insightful comment provided by the reviewer and acknowledge the significance of the possible implications of the crosstalk of TRIM family proteins with NLRP3 inflammasome in lymphomagenesis. We have therefore dedicated a separate section in the manuscript (section 5), entitled “Exploring the crosstalk of Tripartite motif (TRIM) family proteins and NLRP3 inflammasome activation in lymphoma”).
Reviewer 4 Report
Comments and Suggestions for Authors
I have read the paper evaluating the role of NLRP3 inflammasome in lymphoma pathogenesis, and potential therapeutic options aimed at NLRP3 signaling, with great interest. I find the review well written. I have only the suggestion to add the chapter dealing with NLRP3 signaling in lymphoid and myeloid premalignant states (i.e. MGUS, MLUS, CHIP), as these stages are of high interest to potentially treat to prevent full blown malignancy to occur. Otherwise, very nice paper.
Author Response
We sincerely appreciate your valuable feedback and thank you for your positive comments. Following your suggestion we have included a new section in the manuscript (Section 7 “NLRP3 inflammasome signaling in lymphoid premalignant conditions”), reviewing the role of NLRP3 inflammasome in premalignant hematological conditions. We would like to note that we have focused on those of lymphoid origin given that the manuscript aims to focus on lymphoma. Nevertheless, we included a brief report on the interplay of CHIP and inflammasome activation in cells of myeloid origin.
Round 2
Reviewer 3 Report
Comments and Suggestions for Authors
The authors have addressed my suggestion. I would recommend the manuscript for acceptance. Check carefully all the typos and inconsistency if any before finalizing.
Author Response
We would like to thank the reviewer for the suggestion that has undoubtedly added to the scientific quality of the manuscript.
We have carefully checked for typos and inconsistencies.